

**Assessing the influence of soil freeze-thaw cycles on catchment water storage – flux – age interactions**
**using a tracer-aided ecohydrological model**
*Aaron A. Smith[1], Doerthe Tetzlaff[1,2,3], Hjalmar Laudon[4], Marco Maneta[5], Chris Soulsby[3]*
[1]IGB Leibniz Institute of Freshwater Ecology and Inland Fisheries Berlin, Berlin, Germany
[2]Humboldt University Berlin, Berlin, Germany
[3]Northern Rivers Institute, School of Geosciences, University of Aberdeen, UK
[4]Department of Forest Ecology and Management, Swedish University of Agricultural Sciences, Umeå, Sweden
[5]Geosciences Department, University of Montana, Missoula, MT
**Correspondence:** Aaron Smith (smith@igb-berlin.de)
**Abstract**
Ecohydrological models are powerful tools to quantify the effects that independent fluxes may have on catchment storage
dynamics. Here, we adapted the tracer-aided ecohydrological model, EcH$_2$O-iso, for cold regions with the explicit
conceptualisation of dynamic soil freeze-thaw processes. We tested the model at the data-rich Krycklan site in northern Sweden
with multi-criteria calibration using discharge, stream isotopes and soil moisture in 3 nested catchments. We utilized the model's
incorporation of ecohydrological partitioning to evaluate the effect of soil frost on evaporation and transpiration water ages, and
thereby the age of source waters. The simulation of stream discharge, isotopes, and soil moisture variability captured the seasonal
dynamics at all three stream sites and both soil sites, with notable reductions in discharge and soil moisture during the winter
months due to the development of the frost front. Stream isotope simulations reproduced the response to the isotopically-depleted
pulse of spring snowmelt. The soil frost dynamics adequately captured the spatial differences in the freezing-front throughout the
winter period, despite no direct calibration of soil frost to measured soil temperature. The simulated soil frost indicated a maximum
freeze-depth of 0.25 m below forest vegetation. Water ages of evaporation and transpiration reflect the influence of snowmelt-
inputs, with a high proclivity of old water (pre-winter storage) at the beginning of the growing season and a mix of snowmelt and
precipitation (young water) toward the end of the summer. Soil frost had an early season influence of the transpiration water ages,
with water pre-dating the snowpack mainly sustaining vegetation at the start of the growing season. Given the long-term expected
change in the energy-balance of northern climates, the approach presented provides a framework for quantifying the interactions
of ecohydrological fluxes and waters stored in the soil and understanding how these may be impacted in future.
**1. Introduction**
Northern watersheds are sensitive hydrologic sites where a significant proportion of the annual water balance is controlled
by the spring melt period (Kundzewicz et al., 2007) and can thus be key sentinels for detecting climate change impacts (Woo,
2013). Recent data and long-term climate projections indicate a significant increase in warming for extensive areas of boreal forests
currently experiencing low-energy, low-precipitation hydroclimatic regimes (Pearson et al., 2013). The implications of the
anticipated hydrological change in these catchments for water resources and freshwater ecosystems raise substantial concerns,
particularly given the limited number of long-term monitoring sites with high-quality data (Laudon et al., 2018; Tetzlaff et al.,
2015). Within changing northern catchments, with high water loss due to transpiration (~ 48 ± 13%) (Schlesinger and Jasechko,
2014), and significant influence of evapotranspiration (*ET*) fluxes on streamflow (Karlsen et al., 2016a), the long-term
ecohydrological implications of vegetation adaptation, plant water use, and the water sources that sustain growth are crucial to





understand and quantify. Vegetation in boreal regions also exerts strong influences on the energy balance of such catchments, with low leaf area index (LAI) conifer forests and shrubs affecting the surface albedo, snow interception and affecting the timing and duration of the largest input fluxes of water during snowmelt (Gray and Male, 1981). However, the interactions between soil water storage and "green water" fluxes of transpiration and evaporation are poorly constrained in northern regions, and the way in which sources of water from inputs of snowmelt and summer rainfall mix and sustain plant growth is only just beginning to be understood (Sprenger et al., 2018a). Assessment of these interactions in northern catchments is further complicated by large temperature variations, and the resulting stagnation of hydrological processes induced by frequent frozen ground conditions. With increasing temperatures and potential changes to the winter soil freeze-thaw dynamics (e.g. Venäläinen et al., 2001), it is important to establish how these affect current vegetation-soil water interactions to project the implications of future change.

The intricate complexities of changes in the land surface energy balance, temporal changes in sub-surface storage due to frost conditions, and vegetation and soil water usage (transpiration and soil evaporation, respectively), are notoriously challenging to continuously monitor (Maxwell et al., 2019), particularly in northern environments, where site access is typically remote and extreme cold can limit in-situ monitoring devices. In these circumstances, the fusion of sparsely available data with hydrological models is an effective method to quantify water fluxes and storage dynamics at different temporal and spatial scales. While the calibration of such models requires significant hydrometric data inputs, recent work has shown that incorporation of stable isotopes can be an effective tool for constraining the model estimations of storage – flux interactions in the absence of direct in-situ measurements. Such models include (but are not limited to); the STARR (Spatially distributed Tracer-Aided Rainfall-Runoff) model (van Huijgevoort et al., 2016), which was developed for tracer-aided simulations and calibration, and adapted for additional cold-regions processes (Ala Aho et al., 2017a and b; Piovano et al., 2018), CRHM (Cold Regions Hydrologic Model) specific for cold regions (Pomeroy et al., 2007), but not currently using tracers, the isoWATFLOOD model (Stadnyk et al., 2013), which has been used to isolate water fluxes with tracer-aided modelling in large-scale applications in northern regions of Canada, and the EcH$_2$O-iso model (Maneta and Silverman, 2013; Kuppel et al., 2018 a and b), which was developed as a process-based, coupled atmosphere-vegetation-soil energy balance ecohydrologic model, and modified to incorporate isotopic tracers (stable isotopes deuterium and oxygen-18, $\delta^2$H and $\delta^{18}$O, respectively). However, apart from EcH$_2$O-iso, which explicitly conceptualises short-term (diurnal and seasonal) and long-term (growth-related) vegetation dynamics and biomass productivity, most of these existing models were mainly developed with a focus on runoff generation ("blue water" fluxes). Consequently, they have very simplistic representation of vegetation – soil – water interactions, estimating $ET$ by approximating the physical transpiration controls of vegetation (e.g. Penman-Monteith and Priestley-Taylor methods) and partitioning fluxes after estimation of actual $ET$ (Fatichi et al., 2016).

Currently, EcH$_2$O-iso, already incorporates some cold region processes, namely snowpack development, a snowmelt routine, and the influence of temperature effects on vegetation productivity. While the depth of the snowpack is not directly estimated (only snow water equivalent is tracked), the surface energy balance incorporates snowpack heat storage to estimate the warming phase with effective snowmelt timing (Maneta and Silverman, 2013). The model additionally estimates the soil temperatures through multiple soil depths, however, freezing temperature and soil frost development are adaptations that are needed for use in catchments with extensive freezing conditions. Soil freeze-thaw has the potential to significantly influence soil moisture conditions, tracer dynamics, and the magnitude and ages of all water fluxes. The incorporation of tracer dynamics to EcH$_2$O-iso open opportunities to strengthen the evaluation of the model processes (Kuppel et al., 2018b) and permits the use of tracers in calibration (Douinot et al., submitted). Here, our overall aim was to provide a framework for assessing vegetation influences on the hydrology of cold-regions by adapting the EcH$_2$O-iso model and testing it in the intensively monitored Krycklan catchment in northern Sweden. The specific objectives of the study are three-fold; 1) to assess the capability of a spatially distributed, physically-based ecohydrological


model to capture the influence of snow and soil freeze-thaw processes on water storage dynamics, and the resulting flux magnitudes
under different vegetation communities (forest vs mire). 2) To examine the influence of soil frost on the dynamics and age of water
fluxes within the catchment, and 3) provide a generic modelling approach for application to other frost affected catchments. In the
adaptation of EcH2O-iso to cold regions and the assessment of the simulated vegetation-soil water interactions with frost
conditions, we aim to improve the understanding and projecting the future role of vegetation in cold regions hydrology.

## 2. Model description and extensions for this paper

### 2.1 EcH$_2$O-iso model

Recent advances in hydrological modelling have included more explicit process-based conceptualisation of ecohydrological
interactions (Fatichi et al., 2016) and the integration of tracer-based data (Birkel and Soulsby, 2015). The EcH$_2$O model (Maneta
and Silverman, 2013) was developed as an ecohydrological model coupling land-surface energy balance models with a physically-
based hydrologic model. This explicitly includes the dynamics of vegetation growth and vertical and lateral ecohydrological
exchanges.

### 2.1.1 EcH$_2$O energy balance

The energy balance is computed for two-layers, the canopy, and surface. The solution of the energy balance is used to calculate
the available energy reapportionment for transpiration, interception evaporation, soil evaporation, snowmelt, ground heat storage,
and canopy and soil temperature. The canopy energy balance is iteratively solved at each time step until canopy temperature
converges to the estimated value that balances radiative (incoming and outgoing short and long wave radiation), and turbulent
energy fluxes (sensible and latent heat) (Maneta and Silverman, 2013; Kuppel et al., 2018 a and b). Long- and shortwave radiation
transmitted through the canopy to the soil and longwave radiation emitted by the canopy toward the ground drive the surface
energy balance. The surface energy balance components include radiative exchanges (incoming and outgoing short and long-wave
radiation), sensible, latent, and ground heat fluxes, as well as heat storage in the soil and in the snowpack. While the energy balance
apportions energy to each storage (i.e. soil and snowpack), when the snowpack is present, estimated surface temperatures refer to
the snowpack surface and the ground is assumed to be at the temperature of the snowpack, which means that conductive heat
transfer between soil and snowpack is 0 (no thermal gradient). Also, when the snowpack is present latent heat for surface
evaporation is set to 0.

### 2.1.2 EcH$_2$O-iso tracer and water age module

EcH$_2$O has previously been adapted to incorporate the tracking of hydrological tracers including stable isotopes (Kuppel et
al., 2018b) and chloride (Douinot et al., submitted), and adapted to compute estimations of water age in water storage and fluxes.
Isotopic fractionation is simulated in soil water using the Craig-Gordon model (Craig and Gordon, 1965), and tracer mixing is
simulated using an implicit first-order finite difference scheme. Full details of the implementation of the isotopic module are in
Kuppel et al., (2018a). These adaptations do not consider fractionation of snowmelt or open water evaporation. Water ages are
estimated assuming complete mixing in each water storage compartment. Similar to other snowmelt tracer models (eg. Ala-aho et
al., 2017a), the snowmelt ages are defined as the time the snow enters the catchment, rather than the time of melt. This results in
older water estimations during the freshet period and a more complete estimate of the time that water has resided in the catchment.

### 2.2 Soil water freeze-thaw adaptation

Hydrology in cold regions can be greatly affected by the freeze-thaw cycles of soil water during the winter, resulting in
reduced liquid water storage capacity during the spring melt and a restricted capability for infiltration due to the expansion of ice
in pore spaces (Jansson, 1998). The depth of the soil frost can have a large influence on the timing of snowmelt runoff and provide



an estimation of the liquid water available within a soil layer (Carey and Woo, 2005). The Stefan equation is a simple energy
balance approach to estimate the progression of soil water freezing (Jumikis, 1977):

$$\Delta z_f = \left[ \frac{2 k_f \left( T_s - T_f \right)}{\lambda \theta} \right]^{1/2} \tag{1}$$

where $\Delta z_f$ is the change in depth of the frost and is a function of the thermal conductivity of the frozen soil layers between the frost
depth and the soil surface ($k_f$), the soil surface temperature ($T_s$), the temperature of freezing ($T_f$), the latent heat of freezing ($\lambda$), and
the liquid soil moisture ($\theta$). As with previous approaches (Jumikis, 1977; Carey and Woo, 2005), the progression of the soil frost
is estimated by discretizing the total soil depth into smaller layers. Within EcH$_2$O-iso, the sub-surface soil regime is discretized
into three soil layers, layer 1 (near the surface), layer 2, and layer 3 (groundwater to bedrock), to resolve the water balance and
estimate soil moisture. Here, the depths of layer 1, 2, and 3 were used as the layers since they intrinsically incorporate the soil
moisture estimations without additional parameterisation. The thermal conductivity of frost affected layers is dependent on the
moisture content of the soil:

$$k_f(i) = \left( k_{sat} - k_{dry} \right) \cdot \left( \frac{\theta(i)}{\phi(i)} \right) + k_{dry} \tag{2}$$

where $k_f(i)$ is the thermal conductivity of frozen soil in layer $i$, $k_{sat}$ is the thermal conductivity of saturated soil, $k_{dry}$ is the thermal
conductivity of dry soil, $\theta(i)$ is the soil moisture in layer $i$, and $\phi(i)$ is the soil porosity in layer $i$. The saturated thermal conductivity
was estimated from the proportions of soil comprised of ice, liquid water, air, organic material, and mineral soil (Carey and Woo,

30   2005):

$$k_{sat} = \prod_{n=1}^{5} k(j)^{f(j)} \tag{3}$$

where $j$ is the thermal conductivity of each volume proportion, $f$ is the fraction of total soil volume, and $k$ is the thermal
conductivity of volume $j$. Without proportions of soil organic and mineral material, the bulk soil thermal conductivity ($k_{dry}$) is
considered the weighted average of organic and mineral thermal conductivity (only 4 total volumes in Eqn 3). Implementation of
Eqns 1-3 are ideal for EcH$_2$O as the model estimates the parameters ($T_s$, and $\theta$) or includes parameterisation of physical
properties ($\lambda$, $k_{dry}$, $\phi$, $k_{water}$, $k_{air}$), and only requires the addition of the thermal conductivity of ice (2.1W/m/ºC, Waite et al., 2006).
Within EcH$_2$O, the estimation of surface temperature is assumed to be isothermal with the snowpack and conduction through the
snowpack is not considered. However, the surface temperature used within the Stefan equation (Eqn 1) is the surface temperature
below the snowpack. To address the conduction through the snowpack, the estimated surface temperature ($T_{Est}$) was damped
with a single unitless parameter ($D$) such that $T_s = T_{Est} \cdot D$.

40       To account for the reduction of the infiltration rate due to ice, models have previously adjusted the soil hydraulic conductivity

(e.g. Jansson, 1998). Here, the reduction in hydraulic conductivity is estimated using an exponential function:

$$K_{wf} = 10^{fc \cdot F} K_{sat} \tag{4}$$

where $K_{wf}$ is the hydraulic conductivity of the soil influenced by ice, $K_{sat}$ is the saturated hydraulic conductivity of ice-free soils, $fc$
is an ice-impedance parameter, and $F$ is the fraction of frost depth to total soil depth. Equation (4) has two key assumptions: no ice
lenses or frost heaving, and no soil volume expansion due to lower ice density (assumed 920kg/m$^3$ at ice temperature 0-5ºC).

### 2.3 Soil frost volume, depth, and water age

As soil frost progresses through the layers, the proportion of liquid water is assumed to decrease at the same rate as the
proportion of unfrozen soil. Similar to other approaches estimating the moisture content of frost-affected soils (Jansson, 1998), a
minimum liquid soil moisture was retained in all frozen soils. This minimum was assumed to be the residual soil moisture ($\theta_r$),
the minimum moisture content required for evaporation and root-uptake. The change in soil moisture of each layer is estimated:



$$\Delta\theta = (\theta(i) - \theta_r) \cdot \frac{\Delta z_f}{d(i) - d_F(i)} \tag{5}$$

where $\Delta\theta$ is the change in liquid water and ice content, $\theta(i)$ is the initial liquid content in layer i, $\theta_r$ is the residual moisture
content, $d(i)$ is the total depth of layer i, and $d_F(i)$ is the depth of frost in layer $i$. Step-wise estimation of freeze and thaw for each
layer is provided in more detail in Appendix A. The water age of the ice is estimated in a similar way to the liquid water ages of
the soil layers (Kuppel et al., 2018b):

$$V_{res}^{t+\Delta t} A_{res}^{t+\Delta t} - V_{res}^t A_{res}^t = q_{in} A^{t+\Delta t} - q_{out} A_{res}^{t+\Delta t} \tag{6}$$

where t is time, $\Delta t$ is the time-step, $V_{res}$ is the volume of ice in storage, $q_{in}$ is the volume of water from the change in soil moisture
during freeze-up (from Eqn 5), $q_{out}$ is the volume of water from the change in soil moisture during thaw (from. Eqn 5), and A is
the water age (subscripts *res* and *in* are the water ages in storage and inflow, respectively). Similar to the isotope and vegetation
modules in EcH₂O, the frost dynamics (i.e. frost depths and water ages) were implemented as an option within EcH₂O.
*2.4 Isotope snowmelt fractionation*
Isotopic fractionation of snowmelt can have a significant influence on the composition of streams (Ala-aho et al., 2018a).
Previous successful applications of a simple approach equation to estimate the isotopic fractionation of snowmelt at multiple
locations has shown that low-parameterised fractionation models can be used to spatially approximate snowmelt fractionation. One
of the noted limitations of the simple snowmelt fractionation approach used in Ala-aho et al., (2018), is the dependence of the
snowmelt fractionation on the past snowmelt volumes rather than current snowmelt rate. The approach was modified to include
the snowmelt rate with one additional parameter using an exponential function:

$$\delta^2 H_{melt} = \delta^2 H_{pack} - \left( S \cdot exp\left( -S \cdot \left( 1 - \frac{SWE - M}{SWE_{max}} \right) \right) \right) \cdot C \tag{7}$$

where $\delta^2 H_{melt}$ is the isotopic composition of the snowmelt, $\delta^2 H_{pack}$ is the composition of the snowpack at the beginning of the time-
step, SWE is the snow water equivalent at the current time, $SWE_{max}$ is the maximum snow water equivalent before melt, $M$ is the
total volume of snowmelt in the current time-step, $S$ is a slope parameter describing the shape of the exponential change of the
snowmelt fractionation, and $C$ is an amplification factor. Higher values of $S$ (10-20) result in larger early melt fractionation and
limited late melt fractionation, while low values of $S$ result in a lower, but more consistent fractionation throughout the melt period.
The isotopic composition of the snowpack is updated at the end of each time-step.
**3. Data and study site**
*3.1 Study site*
Svartberget (C7, 0.49 km²) is a small subcatchment situated in the headwaters of the Krycklan catchment (64°, 14′N, 19°46′E)
in northern Sweden. Svartberget is a well-studied site with long-term data collection including: streamflow (1991-present), stream
chemistry (2000-present), and hillslope transect measurements (soil moisture and water chemistry). Svartberget has two
subcatchments, Västrabäcken (C2, 0.12 km²) and Mire (C4, 0.18 km²) (Fig. 1). The topographic relief of C7 is 71 m (235 – 306 m
a.s.l.), with 57 m of relief in C2 (247 – 304 m a.s.l.) and only 26 m of relief in C4 (280 – 306 m a.s.l.) (Fig 1). The climate is
subarctic (in the Köppen classification index), with annual precipitation of 614 mm, evapotranspiration (ET) of 303 mm, mean
relative humidity of 82 %, and a 30 year mean annual temperature of 1.8 ºC (Laudon et al., 2013). The relatively low topography
results in no observable influence of elevation on precipitation (Karlsen et al., 2016b). The catchment experiences continuous
snowpack development throughout the winter, accounting for approximately a third of the annual precipitation and lasting on
average 167 days (Laudon and Löfvenius, 2016). The large quantity of snowfall results in a dominant snowmelt-driven freshet
period (Karlsen et al., 2016a). Till (10 – 15 m thick) covers the majority of the downstream catchment area (C7, 92% downstream



of C4) with intermittent shallow soils in the headwaters of C2 (Fig. 1a). The catchment is predominantly forest covered (82% total,
98% downstream of C4), with Scots Pine (*Pinus sylvestris*), Norway Spruce (*Picea abies*), and Birch (*Betula spp.*). The Mire (Fig
1b) is dominated by *Sphagnum* mosses.

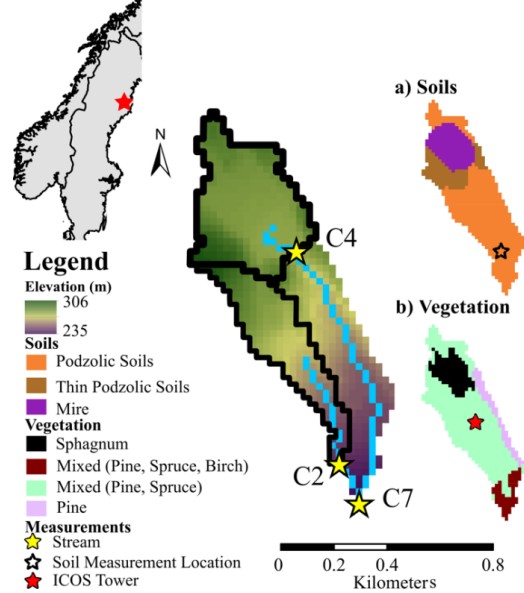

**Figure 1: Location of the Svartberget within Sweden and its elevation profile with the channels and stream measurement locations**
**(yellow). Inset figures show (a) catchment soils, and (b) catchment vegetation.**

## 3.2 Model data

### 3.2.1 Stream discharge and isotope datasets

The discharge at the three streamflow locations has been measured with hourly stream stage measurements using pressure
transducers. V-notch weirs improve measurement accuracy, aided by monthly salt dilution gauging to validate results. Average
discharge in the catchment varies from $9 \times 10^{-4}$ m³/s (C2) to $4 \times 10^{-3}$ m³/s at the outlet (C7), with maximum discharge events up to
0.1 m³/s (C7) during spring freshets (0.02 m³/s and 0.03 m³/s at C2 and C4, respectively). Stable isotopes $\delta^2$H and $\delta^{18}$O
determinations were carried out for samples collected every two weeks at each site. Long-term average $\delta^2$H is similar between
streams (-95.5, -94.5 and -95.6 ‰ for C7, C2, and C4 respectively), with the highest isotopic variability at site C4 (standard
deviation (SD) of 7.9 ‰) and lowest at C2 (SD of 4.5 ‰) with C7 intermediate.

### 3.2.2 Meteorological datasets

Precipitation (rain and snowfall), temperature, wind speed, and relative humidity were measured daily at the Svartberg
meteorological station, 150 m southwest of the catchment. Radiation data, incoming longwave and shortwave radiation, were
obtained at 3-hourly time-steps and 0.75 x 0.75 km grid resolution from ERA-Interim climate reanalysis (Dee et al., 2011). During
the study, a 150 m observation tower (Integrated Carbon observation system, ICOS Tower) was installed within the catchment.
Data from the ICOS tower were available from 2014 to 2015. The ICOS tower measures energy fluxes, latent and sensible heat,
and net radiation, among other atmospheric parameters. The isotopic composition of precipitation was determined on daily bulk
samples following each major rain and snow event. The average precipitation $\delta^2$H (weighted mean -95.1 ‰) is similar to the stream
isotopic composition, though the isotopic variability is between 4.4 – 7.8 times larger.





**Table 1: Datasets used for forcing, calibration and validation within the Svartberg catchment**

| Input Meteorological Forcing Data | | | |
|---|---|---|---|
| Air Temperature (minimum, maximum, and mean) (ºC) | Svartberg | Daily | 2005-2016 |
| Precipitation (m/s) | Svartberg | Daily | 2005-2016 |
| Wind speed | Svartberg | Daily | 2005-2016 |
| Relative Humidity | Svartberg | Daily | 2005-2016 |
| Longwave Radiation | ERA-interm | Daily | 2005-2016 |
| Shortwave Radiation | ERA-interm | Daily | 2005-2016 |
| $\delta^2$H Precipitation | Svartberg | Event-based | 2005-2016 |
| **Calibration and Validation Datasets** | | | |
| | Location | Resolution | Time-Period |
| Discharge | C7 | Hourly | 2005 – 2016 |
| | C2 | Hourly | 2005 – 2016 |
| | C4 | Hourly | 2005 – 2016 |
| Stream isotopes | C7 | Biweekly | 2005 – 2016 |
| | C2 | Biweekly | 2005 – 2016 |
| | C4 | Biweekly | 2005 – 2016 |
| Soil Moisture | S12 | Hourly at depth of 5, 10, 20, 30, 40, 60 cm | 2013 – 2016 |
| | S22 | Hourly at depth of 6, 12, 20, 50, 60, 90 cm | 2013 – 2016 |
| **Validation Only Datasets** | | | |
| | Location | Resolution | Time-Period |
| Soil Isotopes (Lysimeter) | S12 | 9 samples: 10, 20, 30, 40, 60, and 70 cm | 2012 |
| | S22 | 9 samples: 10, 20, 35, 50, 75, and 90 cm | 2012 |
| Soil Isotopes (Bulk Water) | S12 | | 2015 – 2016 |
| | S22 | 7 samples: 10, 20, 30, 40, 60, and 70 cm | 2015 – 2016 |
| Soil Temperature | ICOS Tower | 30 min @ 4 locations at depths 5, 10, 15, 30, and 50 cm | 2014 – 2015 |
| Net Radiation | | 30 min | |
| Latent Heat | | 30 min | |
| Sensible Heat | | 30   in | |


*3.2.3 Soil moisture and isotope datasets*

11        Soil moisture sensors were installed in 1997 and replaced at the beginning for 2013. The soil moisture sensors were installed

at the hillslope transect location at 4, 12, 22, and 28 m locations from the C2 stream. The depths of the soil moisture measurements
slightly differ between sites (Table 1); however, the depths encompass shallow and deep soil waters. Soil sensors have also been
installed in the area surrounding the ICOS tower, measuring soil temperature at 4 locations and 6 depths (10, 20, 30, 40, 60, and
70 cm) (Table 1) which can provide a proxy for the depth of the frost. Soil isotopes ($\delta^2$H and $\delta^{18}$O) were measured at multiple
depths (2.5 cm increments) measured via lysimeters (2012) and bulk water samples (2015 – 2016).
***3.3 Model set-up and calibration***

18        The C7 catchment was defined with a grid resolution of 25 × 25 m² to balance adequate differentiation of multiple locations

on the soil water transect while maintaining computational efficiency. The 25 m grid includes adjacent soil pixels for S12 and S22,
with sites S04 and S28 within the same grids as S12 and S22, respectively. All simulations were conducted on a daily time-step
between January 2005 and September 2016. The period from January 2005 to December 2009 was used as a spin-up period with
measured hydrologic data, to stabilize $\delta^2$H, $\delta^{18}$O composition, and water ages in each of the model storage units. Initial analysis of
the measured discharge from 2000-2016 revealed the highest and lowest annual discharge years were between 2010 and 2014.
Consequently, calibration was carried out for the period between January 2010 and December 2014. The validation set used was
the remaining period from January 2015 – September 2016. Within the biomass module, the vegetation dynamics for leaf growth



and carbon allocation were held at steady state to minimize the parameterisation and focus on the soil freeze-thaw cycles. As
temperature effects and water stress are less sensitive for conifer trees, a relatively constant leaf area index and needle growth/decay
rate were maintained (Liu et al., 2018). Evaporative soil water fractionation was activated using similar parameterisation to Kuppel
et al. (2018b), as this has previously been identified as an influential summer process in the catchment (Ala-aho et al., 2017a). Soil
relative humidity was estimated using Lee and Pielke's (1992) approach, and values of kinematic diffusion were estimated as
presented by Vogt (1976) (0.9877 and 0.9859 for $H^2/H^1$ and $O^{18}/O^{16}$ ratios, respectively). Parameterisation of the model was
conducted for each soil type (3 soil types, Fig 1a) and vegetation type (4 types, Fig 1b).

33        A sensitivity analysis established the most sensitive parameters to be used in calibration using the Morris sensitivity analysis

(Soheir et al., 2014). Parameters were assessed using 10 trajectories using a radial step for evaluating the parameter space. The
parameter sensitivity was evaluated using the mean absolute error. Results of this are shown in Appendix B. Sensitive parameters
were calibrated using Latin Hypercube sampling (McKay et al., 1979) with 150,000 parameter sets and a Monte Carlo simulation
approach to optimize the testing of the model parameter space.
*3.4 Model evaluation*

39        The model output was constrained using measurements of stream discharge (3 sites, Fig. 1), stream $\delta^2H$ (3 sites, Fig. 1), and

soil moisture (2 sites, Fig. 1a). The 8 measurement datasets were combined into a multi-criteria calibration objective function using
the mean absolute error (MAE) with the cumulative distribution functions (CDFs) of the model goodness-of-fit (GOF) (Ala-aho
et al., 2017a; Kuppel et al., 2018 a and b). The MAE moderated over-calibration of peak flow events, typical for functions like the
root mean square error, and Nash-Sutcliffe efficiency, as well as being consistent with previous studies in the region (Ala-aho et
al., 2017a). To focus on the dynamics of soil moisture, given the coarse model grid, measured and simulated values were
normalized by their respective mean values prior to analysis. From the CDF method, the 30 "best" simulations were selected for
evaluation and are presented using 95% spread of predictive uncertainty (Kuppel et al., 2018b). The parameters achieved through
calibration are shown in Appendix C. Model results were verified against the remaining years of discharge, soil moisture, and
stream flow $\delta^2H$, as well as independent time series of soil isotopes (bulk and lysimeter), net radiation, sensible heat, latent heat,
and frost depth (estimated from depth-dependent soil temperatures).

50        The evaluation of changes to water ages due to soil frost was conducted by comparing the ages within the catchment for

simulations of the 30 "best" parameter sets with and without frost. These were conducted without frost by turning frost dynamics
off within the model. Freeze-thaw effects on evaporation and transpiration ages were evaluated as the difference between frost and
non-frost simulations. Positive values indicate older water with the frost while negative values indicate older water with frost-free
simulations. The age differences were only considered on days when both frost and non-frost simulations simulate a flux greater
than 0 mm/day.
**4. Results**
*4.1 Simulation results*

58        Calibration captured dynamics of both high and low flow discharge periods through both the calibration period (2010 – 2014)

and validation period (2015 – 2016), with a maximum mean stream flow MAE of $2\times10^{-3}$ m³/s for C7, and a maximum mean stream
$\delta^2H$ MAE of 5.8 ‰ at C4 (Table 2). Due to extreme high and low flow periods in the calibration period, it was unsurprising that
the resulting MAE was higher than in the validation. The MAE of the soil moisture calibration was also reasonable, with average
MAE of 0.05 and 0.09 for sites S12 and S22, respectively. With the normalization of the soil moisture, the low MAE indicates that
the dynamics in the model correspond well to those measured. The optimization of the GOF for 3 measures (discharge, stream





δ2H, and soil moisture) at 8 locations resulted in a compromise for all streams. Simulations yielded better (lower) MAE for
discharge and isotopes of individual streams.
**Table 2: Calibration and validation efficiency criteria, shown as mean efficiency for all multi-calibration criteria**

|  | Site | Calibration (2010 – 2014) MAE | Validation (2015 – 2016) MAE |
|---|---|---|---|
| Discharge | C7 | $2\times10^{-3}$ m³/s | $6\times10^{-4}$ m³/s |
|  | C2 | $1\times10^{-3}$ m³/s | $1\times10^{-4}$ m³/s |
|  | C4 | $1\times10^{-3}$ m³/s | $3\times10^{-4}$ m³/s |
| $\delta^2$H | C7 | 4.8 ‰ | 4.0 ‰ |
|  | C2 | 4.6 ‰ | 3.8 ‰ |
|  | C4 | 5.8 ‰ | 3.9 ‰ |
| Soil Moisture | S12 | 0.05 | 0.09 |
|  | S22 | 0.09 | 0.09 |
| Latent Heat | ICOS Tower | N/A | 13.1 W/m² |
| Sensible Heat | ICOS Tower | N/A | 29.5 W/m² |
| Net Radiation | ICOS Tower | N/A | 31.0 W/m² |
| Soil Frost Depth | ICOS Tower | N/A | 0.03 m |

Temporal variability of δ²H in each of the streams was captured quite well throughout the calibration and validation periods
(Fig 2 a – c). The largest offsets in modelled isotopic composition occurred during the winter low flow conditions. The simulated
stream isotopes tended to retain a slight "memory" effect from the more enriched late summer. This was likely due to the
underestimation of discharge during winter (Fig 2 d – f) which slowed the flushing of the more enriched water. Overall though,
discharge was adequately simulated for each site, notably during the spring melt and summer months. While flows were
underestimated during the winter, the difference between simulations and measurements were typically $< 1\times10^{-3}$ m³/s. The weight-
median water ages of each of the three streams were broadly similar, 2.8, 2.6, and 3.1 years for C7, C2, and C4, respectively (Fig
2 g – i). These stream ages were generally older than previous estimates, with deeper soil layers and complete mixing in each
compartment tending to increase the average age. The depth of the soil layers in the peat and podzolic areas are the primary drivers
for water age, with a ~1:1 relationship (Appendix D). Water age decreased during the annual freshet, driven by the younger
snowmelt and frozen soil water ages (typically 150 – 200 days old). The rapid runoff during the freshet limited the long-term
influence of the younger water ages on the stream water at each of the sites as older groundwater dominated low flows.





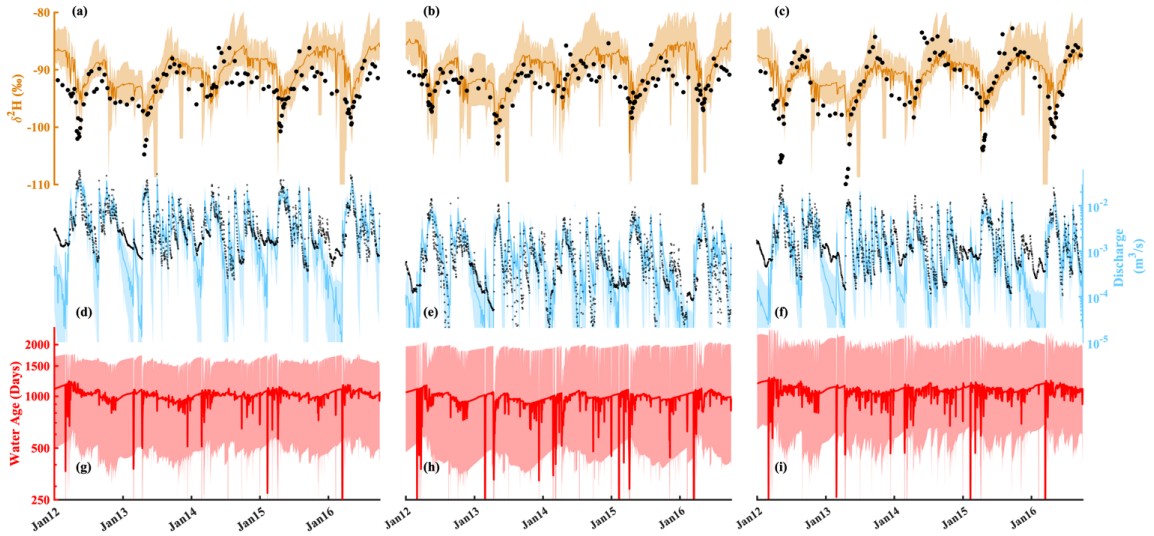

**Figure 2: Calibration 95% maximum and 5% minimum bounds, median simulation (solid line), and measured data (black circles) of δ²H for (a) C7, (b) C2, and (c) C4; discharge for (d) C7, (e), C2, and (f) C4; and stream water age for (g) C7, (h) C2, and (i) C4.**

### 4.2 Soil moisture, isotope, and water ages

Simulated soil water isotopes (note that the model did not use isotopes during calibration) mostly captured those measured in both bulk water (2015 – 2016) and lysimeter water (2012) within the 90% simulation bounds at the S12 and S22 sites (Fig 3 a & b). Isotope dynamics were best captured at site S12, with early season depletion due to snowmelt and enrichment of the previous summer. While most variability was captured within the 90% bounds, the magnitude of the intra-annual contrasts at site S22 was not fully reproduced. Similar to the soil isotopes, dynamics of simulated soil moisture (calibrated) were captured at both S12 and S22, with better simulation performance at S12 (Fig 3 c & d). The model struggled to simultaneously reproduce the more dynamic soil moisture at S12 with the relatively damped soil moisture post-melt at S22 in the adjacent cell under the same soil parameterisation. Rather, the same parameterisation resulted in balancing the conditions observed at S12 and S22. The large declines in measured soil moisture during the winter months were captured with the soil frost module (Fig 3 c & d). The modelled decline in the soil moisture resulted from the transition of soil water from liquid to ice. Water ages in layers 1 and 2 at each site showed noticeable intra-annual variability, and gradually declined during the growing season (May – September) and increased during the winter due to negligible water inflow (Fig 3 e & f). The variability of the soil water ages in layers 1 and 2 was similar, though the ages in layer 2 were significantly older. While S12 is closer to the stream, water ages in S22 were generally older in both layers 1 and 2.





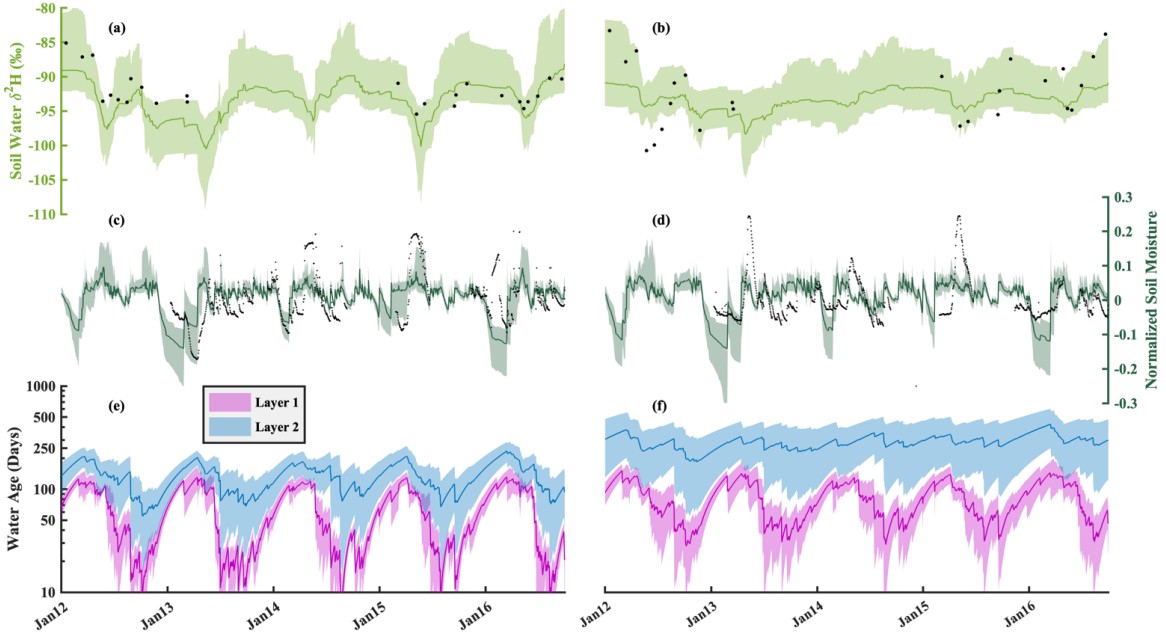

**Figure 3: Simulation 90% bounds and mean simulation (solid line) for the average of layer 1 and 2 δ²H for (a) site S12, and (b) site S22; the average of layer 1 and 2 normalized soil moisture for (a) site S12 and (b) site S22; and water ages of soil layers 1 and 2 for (e) site S12, and (f) site S22.**

### 4.3 Soil freeze-thaw simulations

Simulations of frost depth revealed large inter-annual variability throughout the catchment (Fig 4 a-d), depending on winter temperatures, snowpack depth, and the soil moisture conditions. Wetter conditions in the mire generally show shallower frost depths than the podzolic soils elsewhere in the catchment. Similar soil conditions for the podzolic and thin podzolic soils (Fig 1a) resulted in negligible differences for estimated frost depth. Overall, estimated frost depth was generally limited by the total number of freezing days. Colder winters (larger numbers of freezing degree days) resulted in deeper frost depths for an equivalent snowpack depth (e.g. Fig 4a vs Fig 4c). Conversely, a deeper snowpack (higher maximum SWE) resulted in a shallower simulated frost depth for years with similar temperatures (e.g. Fig 4 a vs c) as the deeper snowpack was a larger storage for incoming radiation. Using 0°C in the soil temperature probes at the ICOS tower as a proxy for the depth of the soil frost, a direct comparison of simulated frost depth and the measured catchment frost depth was completed without calibration. Simulated frost depth showed good agreement with observed 0°C soil temperature depth, imitating the rapid increase in frost depth in 2014 and a more gradual increase in 2015 (Fig 4e). Late winter soil frost depth was estimated to be shallower and varied more rapidly than the observed 0°C soil temperature depth (Fig 4e). The median estimated soil depth against the measured 0°C soil temperature depth showed that estimate soil thaw was too rapid, and thaw completed too early.





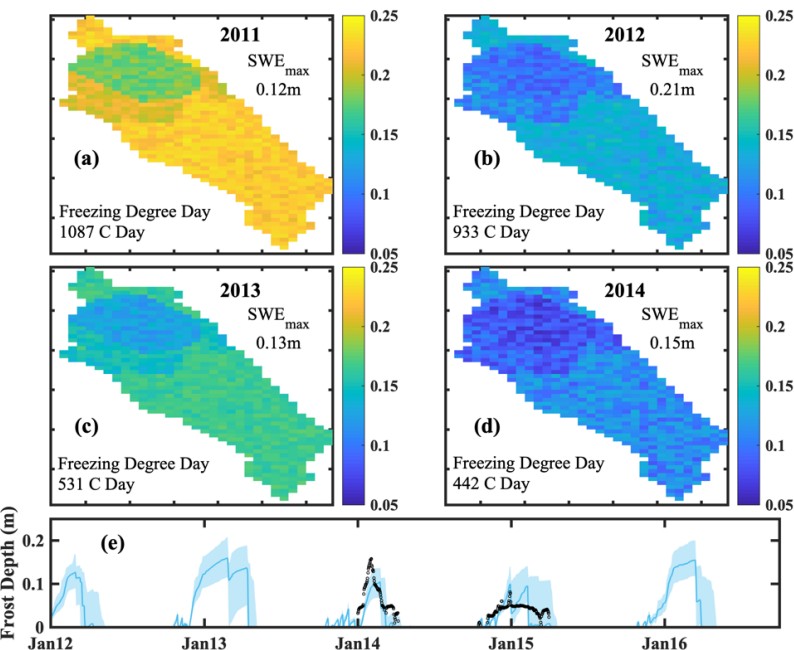

**Figure 4: Mean simulated soil frost depth during the peak soil frost depth in winter (a) 2010-2011, (b) 2011-2012, (c) 2012-2013, and (d) 2013-2014. 90% uncertainty bounds of the simulated frost depth at the ICOS Tower with the depth of the 0°C soil temperature measured at the ICOS Tower (black circles)**

*4.4 Evaporation and transpiration*

While the evaporation, transpiration, and energy balance datasets were not included in the calibration, modelled energy balance
components (sensible heat, latent heat, and net radiation) showed reasonable agreement to observed values in 2014-2016 at the
ICOS Tower. There was an under-estimation of net radiation and sensible heat throughout the growing season (Fig 5 b & c), and
an underestimation of latent heat late in the year (Fig 5a). While the MAE of the latent heat was relatively small (13.1 W/m²)
considering that they were not used for calibration, net radiation and sensible heat had a notable maximum bias (~30 W/m²) during
summer. Simulations of total daily evaporation (soil and interception) and transpiration had a similar pattern, with transpiration
accounting, on average, for 54% of total evapotranspiration. Throughout the year, the simulated proportion of transpiration to total
evapotranspiration ranged from 31 – 72% except for the spring periods (Fig 5d). The late onset of evaporation resulted from the
assumption that soil evaporation was negligible while the snowpack remains, which potentially lead to an under-estimation of
evaporation during the melt.

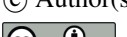



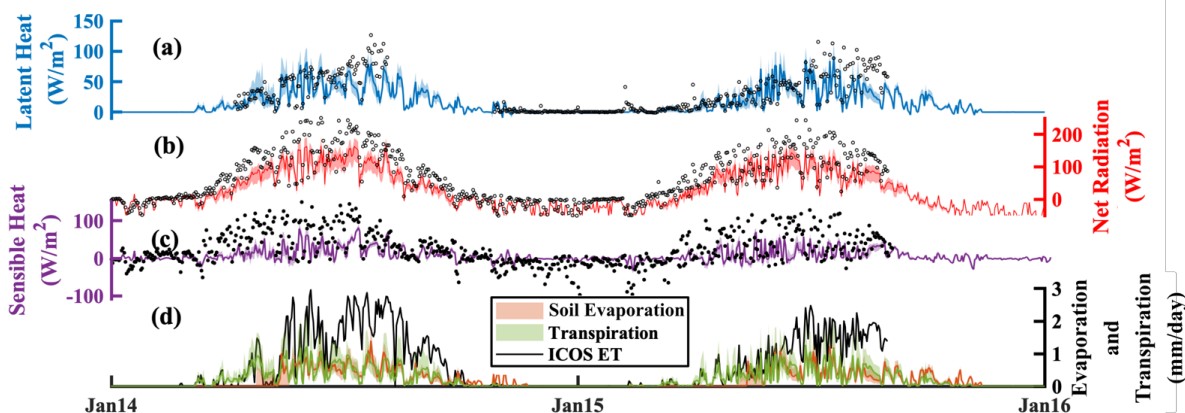

**Figure 5: Energy balance component of (a) estimated latent heat (90% and mean values), (b) estimated net radiation (90% and median values), (c) estimated sensible heat (90% and median values) and (d) estimated soil evaporation and transpiration (90% and mean values), at the ICOS Tower site with the estimated total evapotranspiration from energy fluxes at the ICOS Tower (black circles where data are available).**

Ages of soil evaporation and transpiration decreased throughout the year (Fig 6 a and b), tracing the decline in soil water ages estimated with the addition of precipitation (age of 0 days). Older water present in evaporation and transpiration water at the start of the year was a mixture of the snowmelt water age and frozen water ages (from the previous summer). Spatial differences in evaporation and transpiration ages were evident throughout the catchment; shown by the difference between the forested ICOS tower site (blue, Fig 6 a & b), and the average for shrubs in the mire (green, Fig 6 a & b). The annual flux-weighted median water age of transpiration was $200 \pm 42$ and $141 \pm 40$ days for the ICOS tower and mire, respectively, while evaporation ages were $48 \pm 11$ and $85 \pm 36$ days for the ICOS tower and mire, respectively. Shallower roots of the shrubs resulted in younger transpiration ages than at the ICOS tower and subsequently resulted in older evaporation ages in the mire due to reduced availability of young water.

Differences between the evaporation and transpiration ages were determined by comparing water ages with the soil frost module activated, against those with the frost module deactivated. Generally, including frost in the simulations resulted in older water (water age difference > 0 Fig 6c) for both evaporation and transpiration. Differences in evaporation age were not as pronounced as transpiration ages due to the slight bias of the evaporation timing (always following the snowmelt). Due to the estimated completion of soil thaw prior to the snowmelt period, the difference between the water ages of evaporation with the influence of frozen ground was modest. Rapid flushing of the soil water due to large snowmelt inputs and spring precipitation resulted in a rapid decline in the differences of transpiration water ages. Within the first month of transpiration, the difference for the frost and non-frost simulations were more notable and approached 200 days when frost limited water movement. However, the relatively lower transpiration rates, which occurred during the spring within these simulations, resulted in a moderate effect on the overall annual transpiration water ages. The effects of soil frost on stream water ages showed little effect, with negligible differences given the relatively old water bias in the stream that only shows some flashes of younger water influence (Fig 2 g – i). While the soil frost increased the stream water ages throughout the year, the effect is well within the relatively large uncertainty bounds of the stream water ages.





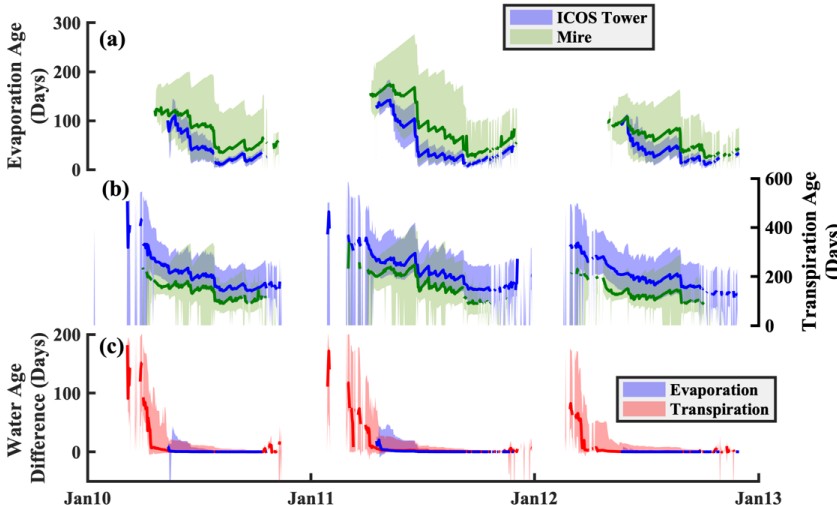

**Figure 6: 90% bounds and median values of the (a) estimated soil evaporation water ages at the ICOS Tower (blue) and in the Mire (green), (b) estimated transpiration water age at the ICOS Tower (blue) and in the Mire (green), and (c) mean difference of evaporation and transpiration water ages when soil frost is not considered.**

## 5. Discussion

### 5.1 Modelling soil freeze-thaw processes in tracer-aided models

Hydrologic models are powerful tools for exploring the internal functioning of catchments, particularly when intensive and long-term monitoring programs are in place to help calibration and testing (Maxwell et al., 2019). Here, the development of a spatially distributed, process-based tracer-aided model for northern climates produced encouraging results reproducing soil frost dynamics despite the model not being directly calibrated to match frost depths observations. The use of streamflows, stream isotope ratios and soil moisture dynamics in calibration proved to be adequate for estimating the dynamics of soil frost depth and timing of the frost onset (Fig 4) and revealed spatial differences in frost depth due to contrasting soil types and moisture conditions. However, there are limitations with the current approach that results in some uncertainty of the effect of soil freeze-thaw on catchment hydrology. To improve the computational efficiency of the model, the temperature of the snowpack was assumed to be isothermal (Maneta and Silverman, 2013), and modified here to include only a single temperature damping parameter. However, snowpacks may have a variable thermal gradient (e.g. Filippa et al., 2014), and is dependent on snow density (e.g. Riche and Schneebeli, 2013), snow surface albedo, wind speed, and liquid water component, among others (USACE, 1956; Meløysund et al., 2007; Sturm et al., 2010). While these additional components may contribute to an improvement in the estimation of soil frost, it likely would not have a significant improvement compared to the simple temperature damping used here with additional calibration to constrain snow water equivalent for more dynamic energy exchange (e.g. Lindström et al, 2002). The simplistic consideration of negligible soil sensible heat storage effects on the soil freeze/thaw processes, consistent with other process-based cold region models (e.g. CHRM, Xie and Gough, 2013), may result in dampened rates of freezing and rapid melting during the spring (Kurylyk and Hayashi, 2016). More delayed melting of the soil frost may have implications for snowmelt runoff, increasing the dynamics of the streamflow isotopic compositions towards more depleted isotopic compositions (Fig 2 a-c). Finally, the simplification of a single soil frost front may have some implications for the snowmelt infiltration to the soil. The single front does not allow for near-surface soil thaw to occur prior to deeper soils and thereby has implications for shallow root-water uptake and evaporation.





Energy fluxes in northern catchments can be highly sensitive to the timing of snowmelt, yielding differences in the surface
and canopy net radiation due to changing albedo and to turbulent fluxes due to alterations in surface temperature. Here, the under-
estimated sensible heat flux during the spring and the growing season could be the result of either the aerodynamic resistance ($r_a$)
to transpiration or an underestimated thermal gradient between the soil and the measured air temperature. Higher estimations in
early season surface temperatures could also result in the shallower, and earlier, simulated soil frost melt relative to the measured
0°C soil temperature depth. While improved timing of the soil-thaw period would likely improve this, direct calibration of the
sensible heat fluxes using the vegetation and soil albedo are likely more effective routes to improved simulations.

### 5.2 Effect of soil freeze-thaw on water ages and implications for northern catchments

The adaptation here of a process-based, spatially distributed model to incorporate some more fundamental aspects of the
hydrology of cold regions provides both the opportunity to improve the representation of key hydrologic functions of cold
catchments, and assess the effect that these additional processes have on transit times and ages of ecohydrological fluxes. While
some work has been conducted on assessing the transit or residence times of ecohydrologic fluxes or their partitioning in northern
(e.g. Sprenger et al., 2018a); however, few studies have included the influence of frozen conditions on the water movement, which
may be significant for the effective transit times during the spring freshet period (Tetzlaff et al., 2018) and flow path modelling in
regions (Laudon et al., 2007; Sterte et al., 2018). Traditionally, water ages in stream water at catchment outlets have been the
primary metrics for assessing the transport of tracers. Here, the relatively old age of stream water, and the under-estimation of soil-
thaw result in only slightly older water ages when soil frost conditions are considered, potentially due to the smaller proportion of
wetland areas (Sterte et al., 2018). The deeper frost depth in the forested regions likely did not reduce the spring infiltration due to
the low moisture content in the soil relative to the more saturated wetlands (Laudon et al., 2007). Additionally, the relatively wide
uncertainty bounds of stream water age estimates present difficulties in assessing the relatively moderate effects of soil frost on
the stream water age (Fig 2). The large dependence of the flows and stream water ages at C7 on the outlet of the large mire at C4
indicates that the water age progressing through the mire will be a strong determinant of long-term change. The flux-weighted
median water age estimations for the streams here were estimated to be substantially older than other tracer-aided hydrologic
models for the catchment (Ala aho et al., 2017a), though were on the upper end of other stream and hillslope transit times from
transit time methods (Peralta-Tapia et al., 2016; Ameli et al., 2017). The reasons for this are largely three-fold. Firstly, the model
was calibrated with soil depths comparable to those observed in the catchment. The calibrated model used soil depths ranging from
1.5 – 6 m, where the shallower soil depths yield stream water ages are comparable to previous studies. Secondly, the complete
mixing assumption within the model does not allow for rapid preferential movement of young water that has been observed in
numerous other recent studies (e.g. Botter et al., 2010; Harman 2015). Incomplete mixing within the model framework would
allow for deeper soil profiles to yield younger water fluxes, as estimated from isotopes alone, albeit at the expense of additional
parameterisation. Lastly, the previous transit time estimations (Peralta-Tapia et al., 2016; Ameli et al., 2017) do not account for
older water ages of the snowpack, or the immobility and aging of frozen soil water, which would increase the estimated water ages.
Unlike stream or soil water ages, low uncertainty of transpiration and soil evaporation ages helps bring new understanding to
how soil frost affects the source contributions of these ecohydrological fluxes which were the focus of the study. Ages of both
transpiration and soil evaporation are consistent with soil profile modelling conducted in the region using the SWIS model
(Sprenger et al., 2018b). However, the dynamics of the age variation are notably different due to the differences in the input water,
where the snowmelt input to the SWIS model assumes a water age of 0 days and does not account for the "green" water fluxes
during the spring months. While the transpiration ages show notable differences when frost, and the corresponding discontinuity
of transit times, is included in the simulation, the evaporation water ages are not greatly affected. The differences are reduced for
both fluxes due to a few potential reasons. Firstly, the timing of the soil thaw has a significant influence on age estimation of soil

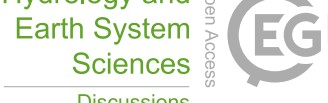



water available for both evaporation and root-uptake. While the general timing and magnitude of the soil frost depth development
seems appropriately captured by the model, even without calibration, soil thaw in late winter was simulated faster than observations
(Fig 4e). There are notable differences between the ages of soil water, soil frost, and the snowpack, where soil frost is representative
of the previous fall soil water, soil water is a younger water mix of the fall soil water and newer precipitation (e.g. from rain-on-
snow and early spring snowmelt), and snowpack is the amount weighed age of solid precipitation. Here, shallower soil frost and
early melting of soil frost in the spring results in step-wise mixing, firstly of soil frost (oldest water) and soil water (moderate age),
then of the soil water mixture and snowmelt (youngest water). Since evaporation, and its corresponding age, only begins following
the end of snowmelt, the greater degree of mixing of soil frost ages with the soil water and snowmelt reduce the influence of the
soil frost on the evaporation ages. Delaying the simulated timing of soil thaw would result in a larger influence of the soil water
ages on both the evaporation and root-uptake.
While the influence of soil frost on stream water ages was limited in this catchment, the results have potentially significant
implications for modelling other catchments with frozen soils. The effect on water ages will likely be the greatest in catchments
where winter precipitation is limited, allowing the soil frost depth to increase from the surface, delaying the soil thaw until after
the primary snowmelt. For evaporation and transpiration water ages, notable spatial differences highlight an essential consideration
for northern climates in the influence of vegetation-type on the source of water fluxes. In many northern areas, past glaciation
results in significant wetlands typically dominated with shrub and herbaceous vegetation. Reductions in soil frost will result in
greater water availability throughout the year, aiding in vegetation growth (Woo, 2013). With the dominance of shallow rooting
profiles in short vegetation and their dependence on younger water, it is likely that the shrub-covered regions of the boreal
catchments will increase in their water usage, and increase the age of soil water and catchment outflows. Finally, the timing of the
evaporation and root-uptake needs to be strongly considered, at both seasonal and diurnal time scales. Soil frost had a strong
influence on the timing of evaporation and transpiration, where the magnitude of both fluxes was greater in simulations without
soil frost and timing of the root-uptake and soil evaporation was delayed due to ice-restricted pore spaces. While such changes are
anticipated, many studies have focused on plot scale studies and with estimated long-term reductions of soil frost depth, larger
scale estimations of these differences are essential to understanding how catchment ecosystems will respond.

## 6. Conclusion

In northern environments, with a rapidly changing climate, quantitative evaluation of vegetation interactions with catchment
soil water is crucial for understanding and projecting catchment responses. The process-based evaluation here of a well-monitored,
long-term study catchment in the northern boreal forest region using a tracer-aided, surface-atmosphere energy-balance model has
provided significant insights into the importance of soil freeze-thaw processes. Tracers were used, not only as a calibration tool,
but as validation metrics, and highlighted the effectiveness multi-criteria calibration of a model at nested scales using discharge,
isotopes, and soil moisture to constrain additional, un-measured, features (e.g. soil frost depth). The progressively younger ages of
evaporation and transpiration throughout the growing season show the dependence of both "green water" fluxes on spring
snowmelt, which remains in soil water towards the end of the growing season. Adaptation of the EcH$_2$O-iso model provided an
opportunity to examine spatial patterns of frost depth throughout the catchment and its ecohydrological influence. Soil frost
responded to both lower winter temperatures (increasing frost depths) and greater snowpack depth (decreasing frost depth). While
there was little influence on the overall timing of water movement at the catchment scale as stream water ages, the greatest influence
was observed within the ecohydrological partitioning, notably with the transpiration ages. Soil frost delays the onset of vegetation
growth and soil evaporation, resulting in older soil water from the previous autumn to sustain early-season transpiration rather than
younger snowmelt. With the implications of reduced numbers of cold days (Guttorp and Xu, 2011), and the dependence of





vegetation growth on the summer temperatures (Schöne et al., 2004) in northern latitudes, this assessment of ecohydrological
partitioning is timely in understanding the effect of climatic change.

**Acknowledgements**
This work was funded by the European Research Council (project GA 335910 VeWa). Marco Maneta recognises funding for
model development and applications from the US National Science Foundation (project GSS 1461576 . The work in the
Krycklan catchment is funded by Swedish Research Council (SITES), SKB, Formas and the KAW program Branch-Points.

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

beginning of each time-step, $k_{fi}$ was estimated for each soil layer. With the thermal conductivity and the depth of frost in each
layer ($z_{fi}$, and conversely unfrozen depth, $\Delta x$), the thermal resistance of each layer was estimated ($R = z_{fi}/k_{fi}$). The depth of
frost in each layer has a maximum of the total depth of the layer. If the total resistance to freeze soil layer $m$ is exceeded (i.e. $T_s >$
$N(m) = \lambda\theta\Delta x(\sum_{j=1}^{m-1} R(j))$, Carey and Woo, 2005), the soil layer is completely frozen. If the total resistance is not exceeded
then partial freezing/thawing of the soil layer occurs:

$$\Delta z_f = -k_f(m) \sum_{j=1}^{m-1} R(j) + \sqrt{k_f(m)^2 \left(\sum_{j=1}^{m-1} R(j)\right)^2 + \frac{2k_f(m)N}{\lambda\theta}} \qquad \text{A.1}$$

where the sum of the terms R(j) are the resistance of frozen soil layers up to the soil layer currently undergoing freezing, and $\Delta z_f$
is the increase/decrease in the depth of frost. The process is repeated each day to estimate the increase (freezing) or decreasing
(thawing) of the frost front.
The limitation of a single, non-depth dependent porosity and hydraulic conductivity in the model structure was modified as
a minor model structure change by using a simple exponential function to decrease the soil porosity and hydraulic conductivity
with depth, similar to the approach employed by Kuppel et al. (2018) for the rooting profile. Similar to the original model
structure, the approach assumes that the soil properties are the same for all soil layers (3), while increased consolidation of soils
at deeper depths result in lower porosity and lower hydraulic conductivity. The porosity is estimated for layer 1 as:

$$\phi_1 = k \cdot \phi \cdot \left(1 - \exp\left(-\frac{d_1}{k}\right)\right)/d_1 \qquad \text{A.2}$$

where $d_1$ is the soil depth of layer 1, $\phi_o$ is the porosity at the soil surface, and $k$ is the exponential rate of change of the porosity.
For layer 2:

$$\phi_2 = k \cdot \phi \cdot \left(\exp\left(-\frac{d_1}{k}\right) - \exp\left(-\frac{(d_1 + d_2)}{k}\right)\right)/d_2 \qquad \text{A.3}$$

where $d_2$ is the depth of layer 2, and for layer 3:

$$\phi_3 = k \cdot \phi \cdot \left(\exp\left(-\frac{d_1 + d_2}{k}\right) - \exp\left(-\frac{d}{k}\right)\right)/(d_1 + d_2) \qquad \text{A.4}$$

where d is the total depth of the soil, the sum of all soil layer depths. Equations A.2-A.4 can also be used to solve for the
hydraulic conductivity in each soil layer (replace $\phi$ with $K_h$ in each equation). The parameterization of $k$ for both the porosity and
hydraulic conductivity with depth should be carefully chosen. Parameterization should always be checked to ensure that the
porosity in layer 3 is greater than the residual soil moisture ($\theta_r$) and the permanent wilting point ($\theta_w$).
*Appendix B: Morris sensitivity analysis*
The aid with parameterization and reduce the total number of parameters used within calibration of the EcH2O-iso model, a
sensitivity analysis with the Morris Sensitivity method and mean absolute error was used to assess how sensitive the model
parameters were to stream discharge, stream isotopes ($\delta^2H$), and soil moisture (calibration time-series). Since the vegetation
dynamics (carbon allocation mechanisms and vegetation growth) were not activated for the calibration, most vegetation
parameters were not included in the sensitivity analysis as they would not result in any changes to observable metrics. These





parameters include: leaf allocation parameters, canopy quantum efficiency parameters, cold stress, and moisture stress leaf
turnover parameters. Parameter sensitivity was assessed using the radial step method and 50 different trajectories. Initial
parameterizations were established using Latin Hypercube Sampling (LHS) to maximize the distance between the randomized
trajectories. Each radial step deviated from the initial parameterization by progressively changing each parameter by half of the
parameters range. For example, initial parameter value of 0.1 with a range of 2 $(0-2)$ results in a new parameter value of 1.1
$(0.1 + (2-0)/2)$. The sensitivity of the parameter was assessed against the original parameterization for the trajectory using the
mean absolute error.

| | | Frost Depth | Soil Age | VMC | Stream d2H | Stream Age | Streamflow |
|---|---|---|---|---|---|---|---|
| **Soil Parameters** | Anisotropy | 0.0 | 0.1 | 0.0 | 0.4 | 0.4 | 0.5 |
| | BC lambda | 0.2 | 0.3 | 0.4 | 0.2 | 0.3 | 0.2 |
| | GW Seepage | 0.0 | 0.0 | 0.0 | 0.3 | 0.2 | 0.3 |
| | Snow T damping | 1.0 | 0.0 | 0.1 | 0.0 | 0.0 | 0.1 |
| | Snowmelt amplification parameter | 0.0 | 0.0 | 0.0 | 0.5 | 0.0 | 0.0 |
| | Layer 1 depth | 0.6 | 0.1 | 0.3 | 0.2 | 0.1 | 0.1 |
| | Layer 2 depth | 0.1 | 0.2 | 0.1 | 0.0 | 0.1 | 0.0 |
| | Layer 3 depth | 0.0 | 0.7 | 0.1 | 0.3 | 0.9 | 0.2 |
| | Ice impedence parameter | 0.0 | 0.0 | 0.0 | 0.0 | 0.0 | 0.0 |
| | Hydraulic Conductivity | 0.1 | 1.0 | 0.4 | 1.0 | 1.0 | 1.0 |
| | Exponential Depth Kh | 0.0 | 0.1 | 0.0 | 0.1 | 0.1 | 0.1 |
| | Leakance | 0.0 | 0.3 | 0.1 | 0.7 | 0.6 | 0.7 |
| | Mannings n | 0.0 | 0.0 | 0.0 | 0.5 | 0.4 | 0.4 |
| | Porosity | 0.6 | 0.2 | 1.0 | 0.1 | 0.3 | 0.1 |
| | Porosity change with depth | 0.0 | 0.1 | 0.1 | 0.0 | 0.1 | 0.0 |
| | Psi AE | 0.2 | 0.2 | 0.3 | 0.2 | 0.2 | 0.2 |
| | Soil heat capacity | 0.0 | 0.0 | 0.0 | 0.0 | 0.0 | 0.0 |
| | Snow fractionation slope parameter | 0.0 | 0.0 | 0.0 | 0.5 | 0.0 | 0.0 |
| | Residual soil moisture | 0.1 | 0.0 | 0.0 | 0.0 | 0.0 | 0.1 |
| | Snowmelt coefficient | 0.5 | 0.0 | 0.1 | 0.2 | 0.1 | 0.3 |
| | Soil thermal conductivity | 0.7 | 0.0 | 0.1 | 0.0 | 0.0 | 0.0 |
| | Soil damping temperature | 0.0 | 0.0 | 0.0 | 0.0 | 0.0 | 0.0 |
| **Veggetation Parameters** | Albedo | 0.0 | 0.0 | 0.0 | 0.0 | 0.0 | 0.0 |
| | Canopy water storage (max) | 0.2 | 0.2 | 0.1 | 0.2 | 0.3 | 0.4 |
| | Emissivity | 0.1 | 0.0 | 0.0 | 0.1 | 0.0 | 0.1 |
| | Gs(light) | 0.0 | 0.0 | 0.0 | 0.0 | 0.0 | 0.0 |
| | Gs(max) | 0.0 | 0.1 | 0.0 | 0.0 | 0.0 | 0.1 |
| | Gs(vpd) | 0.0 | 0.1 | 0.0 | 0.0 | 0.0 | 0.0 |
| | KBeers | 0.0 | 0.0 | 0.0 | 0.0 | 0.0 | 0.0 |
| | LAI | 0.1 | 0.1 | 0.0 | 0.1 | 0.1 | 0.7 |
| | LWPc | 0.0 | 0.0 | 0.0 | 0.0 | 0.0 | 0.0 |
| | LWPd | 0.0 | 0.0 | 0.0 | 0.0 | 0.0 | 0.0 |
| | T(max) | 0.0 | 0.0 | 0.0 | 0.0 | 0.0 | 0.0 |
| | T(min) | 0.0 | 0.0 | 0.0 | 0.0 | 0.0 | 0.1 |
| | T(opt) | 0.0 | 0.0 | 0.0 | 0.0 | 0.0 | 0.1 |

*Normalized Sensitivity* (color scale: 0 to 1)


**Figure B.1.: Normalized mean absolute error for each time-series. Values of 1 indicate the most sensitive parameters and 0 indicates the least sensitive parameters. Additional information on the naming convention is found at https://ech2o-iso.readthedocs.io/en/latest/Setup.html**

Mean absolute error was averaged for all trajectories to determine the mean sensitivity of each parameter. For streamflow, the
parameters most sensitive are a mixture of soil parameters (e.g. hydraulic conductivity, snowmelt coefficient, and anisotropy),
channel parameters (e.g. Mannings n and leakance), and vegetation parameters (e.g. leaf area index and canopy water storage).
Stream isotopes are similarly affected by soil and stream parameters, and show significant influence of the newly introduced





snowmelt fractionation parameters (slope and amplification parameters). As anticipated, soil moisture simulations are most
sensitive to soil parameters, with the most sensitivity related to the porosity. The parameter sensitivity of frost depth was also
tested due to the newly implemented frost module. The frost depth was most sensitive to the snow temperature damping
parameter, with other sensitivity related to winter and thermal processes (snowmelt coefficient and soil thermal conductivity).
Since it is not possible to directly calibrate the soil water or streamflow water ages, the sensitivity analysis was evaluated to
provide additional assessment of which parameterizations will result in changes and uncertainties of the water ages. Except for
the soil heat capacity (set to $2.205 \times 10^6$ W kg$^{-1}$ C$^{-1}$), residual soil moisture (set to 0.05), and the temperature at the damping depth
(set to 5$^o$C), all other soil parameters (Fig B.1) were used in calibration since they showed to be sensitive for the calibration time-
series. Vegetation parameters used in calibration included the canopy water storage, leaf area index, maximum stomatal
conductance ($G_{s,max}$) and two soil-based vegetation parameters controlling the sensitivity of vegetation to suction potential and
moisture content.
*Appendix C: Model calibration parameters*
Model calibration showed a reduction in the parameter space for almost all parameters, where the maximum range of parameters
is shown with the upper and lower bounds of the plots (Fig C.1). Differences between parameterization of soils was most
noticeable for anisotropy, hydraulic conductivity ($K_h$) and porosity ($\phi$), while for vegetation, canopy storage, maximum stomatal
conductance ($G_{s,max}$) and leaf area index (LAI) varied most between the vegetation types.





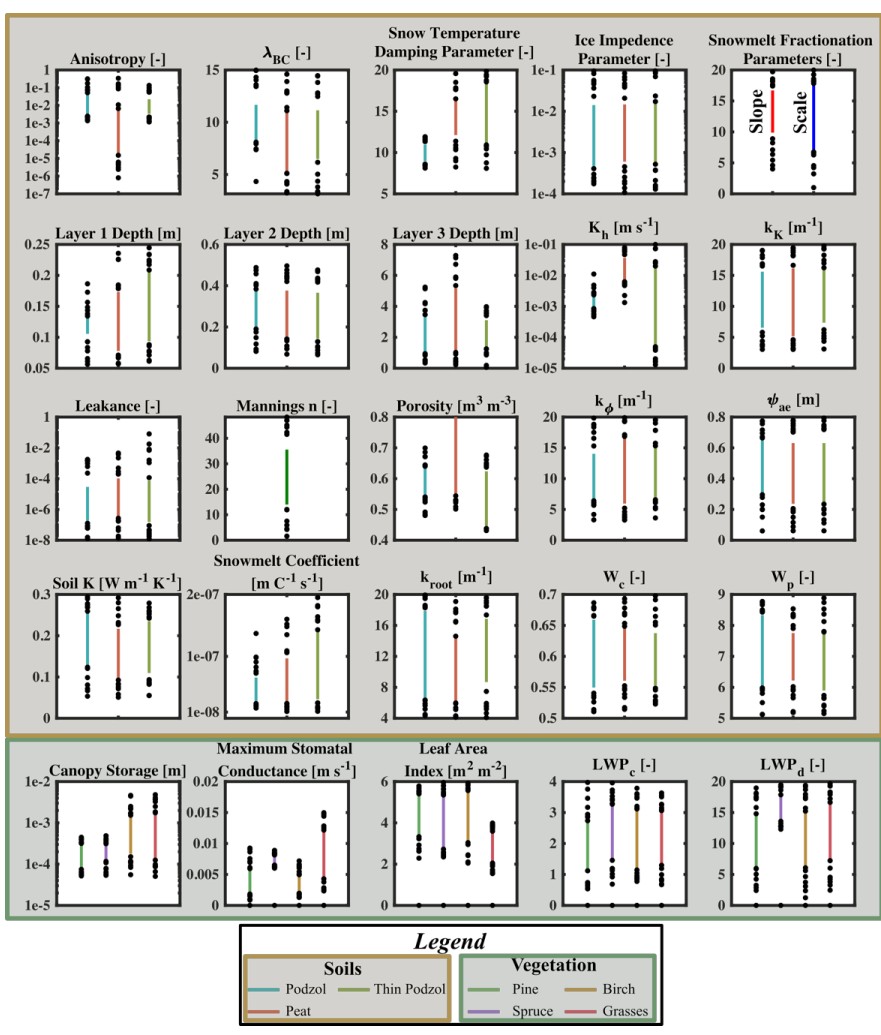


**Figure C.1: Calibration parameterization for soil and vegetation parameters for each soil and vegetation type. Lines represent the 25-**
**75th quantiles and circles are outliers for the quartiles.**
*Appendix D: Simulated layer depth vs. stream water ages*
Stream water ages were expected to predominantly controlled by the depth of the third soil layer in each of the soil types (Fig
B.1). The relatively old stream water ages observed within the simulations were on the upper end of the previously simulated
water ages, with median ages of ~3 years for all streams. A direct correlation was observed for each stream to the depth of the
dominant soil type of the sub-catchment. The overall catchment outlet (C7) was dominated by podzolic soils, and showed a
strong (0.82 correlation coefficient) 1:1 relationship with a regression of the stream water age to the depth of the soil water (1
year of stream age for each meter of soil depth) (Fig D.1.). A similar, stronger, 1:1 relationship of stream water age to soil depth
was observed at the outlet of sub-catchment C2, which was also dominated by podzolic soils (Fig D.1). Unsurprisingly, in the
peat dominated C4 catchment, a strong relationship (0.86 correlation coefficient) was observed between the peat soil depth and
stream water age; however, the stream water became older with soil depth than in the podzolic soil dominated catchments.



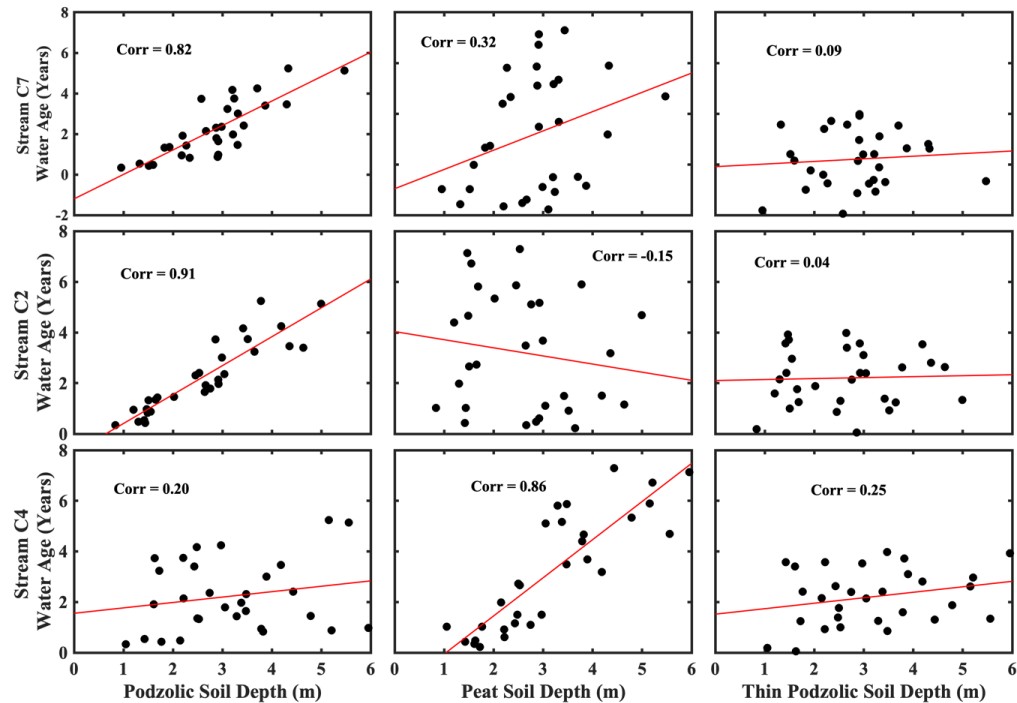


**Figure D.1: Linear regression and correlation coefficients for stream water age at each sub-catchment against the soil depth**