# Peer review of "Assessing the influence of soil freeze-thaw cycles on catchment water storage – flux – age interactions using a tracer-aided ecohydrological model"

_Hydrology and Earth System Sciences, 2019_

## Referee Comment (RC1) · Christopher Spence (Referee) · 10 Apr 2019

In this paper, the authors apply a tracer aided hydroecological model to assess the role of frozen ground on water fluxes, storage and ages in a cold regions watershed in northern Sweden. The model performed well enough to make sound conclusions about the relative magnitude of fluxes and the distribution of ages of water comprising different components of the water budget. The subject matter of this research is very relevant in regards to beginning to address larger questions about how climate, vegetation and hydrology interact. These are important questions as the globe warms,
and tools such as the model introduced here will be important for predicting and attributing change. The paper is well written. I have some minor suggestions where improvements could be made. A bigger concern is an incomplete explanation of how the authors assessed the role of ground frost on water fluxes and ages. The authors explain that they turn frost dynamics off in the model to do so. I perhaps misunderstand, but how is it possible to not have the soil freeze if the same forcing dataset is used? This is a crucial piece in the methodology and it needs better explaining than currently exists. Without it, the paper does not achieve its goals.

There are some suggestions I have that might improve the presentation. My specific comments are below.

Page 1 Line 34: It is not clear how the limited number of monitoring sites is tied to implications of hydrological change. Maybe rephrase to "The limited number of long-term monitoring sites with high quality data is a concern because it may prove difficult to document the anticipated hydrological change in these catchments".

Page 4 Line 39: How is the equation presented here related to the assumption that the ground and snowpack temperature are the same?

Page 5 Line 50: Here and elsewhere, the paper would benefit greatly from the inclusion of units when introducing variables. Page 5 Line 50: These equations imply the soil moisture scheme assumes no movement of water in the column? I cannot think this is correct, and I must misunderstand. Could the authors please improve the clarity here?

Equation 6: It might be the version I see, but the equation seems incomplete and the description doesn't quite match with no mention of outflow.

Page 5 Line 62: Perhaps show the equation from Ala-aho, to show the difference to the reader.

Figure 1 could be better drafted and explicitly label the locations of S12 and S22.

Page 6 Line 90: Not all of this section includes model data, and some is observational

data. You could perhaps retitle the section "Observations".

Page 7 Line 20: Perhaps put the simulation period right at the beginning of the section. Figure 2: Could the authors add a sentence or two explaining why the water ages bottom out every now and then? Perhaps I have missed it.

Page 10 Line 89: Are the words dynamic and damped mixed up?

Figure 3: Please explain what 'normalized' means.

Figure 3: Also, why does the soil water age get younger as the summer progress? The paper would benefit from a few sentences explaining this behaviour.

Figure 5: Just so apples are compared to apples, perhaps total modelled evaporation and transpiration so that it can be more easily compared to the ICOS data.

Page 13 Line 41: A citation might be useful here because the data from this paper do not support such a statement.

Page 15 Line 88: The authors have access to soil temperature data that could show if this is underestimated. A figure might help address this gap. Also, please explain how the assumption of no temperature gradient through the snowpack influence these results.

Page 15 Line 93 – 99: There are some typos through this section that could be fixed.

Page 16 Line 27: I missed where the ages of the soil frost are provided. It would be valuable to show them.

Page 16 Line 30: It would be helpful to provide data on the relative values of these fluxes and storages in the text here to let the reader know how important each is to determining the age of water.

Page 16 Line 32: Maybe rephrase to "…of older soil frost with younger soil water and snowmelt reduces….."

Page 16 Line 35: Was it limited or just hard to detect within the uncertainties of the model? This is an important point of discussion that is missing.

Page 16 Line 43: I am not convinced the results of the research support these statements. Please clarify. If more water is pulled from soil subject to warming would not that speed up the pattern observed in Figure 3? And in turn reduce age?

---

## Referee Comment (RC2) · Anonymous Referee #2 · 13 Apr 2019

The paper by Smith et al., seeks to use a previously developed ecohydrological model (EcH2O-iso) to further understand the partitioning, water storage, flux and age inter-actions, particularly in the context of cold, northern catchments. This novelty of this contribution is that they have adapted the model to include soil freezing, and the impact of soil freezing on water ages. As the authors note, most model estimations of storage-flux interactions oversimplify vegetation-soil-water interactions, while EcH2O-iso provides a generic and relatively simplistic (in some parts) modeling approach to evaluate storage and water ages in cold environments. The model of course has many

limitations related to the process physics and the assumption of complete isotope mixing within each compartment, which may not hold true. However, the authors are transparent as to its shortcomings in most places, and it is of little value to be overly picky with regards to the choices that are made. The manuscript is well written, and the figures are clear and of high quality. I would like the authors to consider the comments below and I believe the manuscript is suitable for publication after minor revisions.

The main conclusion of the work is that soil frost had an early season influence on the ages of transpiration, with less of an influence on water ages of evaporation. Second, that the new module can simulate soil frost dynamics. While I do not dispute this, it is unsurprising that the Stefan-type of equations can simulate frost well, this approach has been used for ages and ages and while perhaps not always a physical realistic representation of ground freezing, it simply works well (as it does here). It would be good for the authors to indicate whey they did not use a more complex thermal scheme, or reference ones. Obviously one would need more soil layers and computational resources would go through the roof, but a bit more on the 'why' this method was used is good.

I would like to focus my comments around the central conclusion re: soil frost and water ages. It would be useful to outline how evaporation and transpiration are partitioned as this would help the reader (although it is likely presented elsewhere) and goes to the central conclusion.

~Equation 1 simulates the depth of the freezing front, but not the soil temperature. I am curious as to how the model simulates soil temperature. I THINK I understand how the surface temperature is driven, and the authors acknowledge that the thermal routine of the snowpack is simple for various reasons. What I'm trying to get at is: does the model simulate a soil temperature and how does this relate to the position of the zero-degree isotherm. Yes, soils will be identified as frozen or unfrozen base on Eq 1, yet is there a modeled soil temperature that simply has no freezing routine? More clarity is needed.

~The central conclusion that soil freezing affects transpiration is fine, but is it simply because the plants are not 'on' when the soil is frozen and soil evaporation is impeded (it certainly would be). When the module is off, plants can transpire, and soils evaporate? Is it this simple? I'm just not sure. More clarity on what drives the transpiration would be helpful as I'm unsure if there can be no transpiration when the rooting zone is frozen – how does this all work?

~Is there sublimation in the model? I see that latent heat is set at 0 when there is snow – why? What impact does this have when snow is melting and sublimation may be important.

~For Equation 7, what is the basis of the amplification factor C. Does equation 7 preserve an isotope mass balance throughout all time steps (I'm assuming so – but it should be stated).

~The authors use ERA-Interim data to drive the radiative component of the model. For a few years, there was overlap. Did they investigate the bias of the ERA data and correct? I'm assuming ERA-I would work well in this location of Europe, but it's good to check as it can have biases which will propagate through the energy balance calculations. The underestimation in net radiation is a bit concerning – and latent heat as well. So after all this, my question is that if latent heat is in fact greater than simulated, what influence would this have on the age estimates (if any?). I assume some and this should be noted.

~On line 79, I'm not sure that the CRHM reference is correct and the Xie and Gough paper describes the thermal routine that is later incorporated into CRHM (see papers by Krogh for example). The XG method is in CRHM, but this is just slightly incorrect referencing.

~The discussion after line 85 is a bit selective and there are dozens of possible reasons for model errors in turbulent fluxes. First the authors state sensible heat fluxes are underestimated but only show latent fluxes so the discussion should be there or

sensible heat data should be provided. Another reason not stated (and noted above) is the nature of the ERA-I data. I'm also unsure as to how snow processes are incorporated into the canopy module re: unloading, albedo change, etc. All I'm saying is that there are many many reasons here where the model could be improved with physics, and avoid suggesting 'direct calibration' is the best way to improve simulations.

~Figures that highlight the differences between soil moisture at depth would be helpful. A few small typos or unclear statements are outlined below:

~Line 80: under different vegetation communities (forest vs mire). 2) To examine the influence of soil frost on the dynamics and age of water (Comma instead of period after (forest vs mire)

~Line 54: qin → subscript needs to be added

~Line 73: comma needed within coordinates

~Line 95: "Stable isotopes determinations were carried out" → Fix wording

~Table 1: Units of precipitation say m/s → should be moved to wind speed. Units need to be added to other dat. "30 min for Sensible Heat says " 30 in" . Column heading needs to say "Time Period" for top row.

~Line 69: stream isotopes tended to retain a slight "memory" effect from the more enriched late summer. . . "contributions"? "water"? I think a word is needed here?

~Beginning Line 95: While some work has been conducted on assessing the transit or residence times of ecohydrologic fluxes or their partitioning in northern (e.g. Sprenger et al., 2018a); however, few studies have included the influence of frozen conditions on the water movement, which may be significant for the effective transit times during the spring freshet period (Tetzlaff et al., 2018) and flow path modelling in "cold" regions (Laudon et al., 2007; Sterte et al., 2018).

~Line 99: Traditionally, water ages in stream water at catchment outlets have been the

primary metrics for assessing the transport of tracers. Should this read: Traditionally, isotopic tracers in stream water at catchment outlets have been the primary metrics for assessing water ages.

~Line 29: snow and early spring snowmelt), and snowpack is the amount "weighed" age of solid precipitation (*Should this be weighted)
* * *

---

## Author Comment (AC1) · 29 May 2019

**General Comments**

In this paper, the authors apply a tracer aided hydroecological model to assess the role of frozen ground on water fluxes, storage and ages in a cold regions watershed in northern Sweden. The model performed well enough to make sound conclusions about the relative magnitude of fluxes and the distribution of ages of water comprising different components of the water budget. The subject matter of this research is very relevant

in regards to beginning to address larger questions about how climate, vegetation and hydrology interact. These are important questions as the globe warms, and tools such as the model introduced here will be important for predicting and attributing change. The paper is well written. I have some minor suggestions where improvements could be made. A bigger concern is an incomplete explanation of how the authors assessed the role of ground frost on water fluxes and ages. The authors explain that they turn frost dynamics off in the model to do so. I perhaps misunderstand, but how is it possible to not have the soil freeze if the same forcing dataset is used? This is a crucial piece in the methodology and it needs better explaining than currently exists. Without it, the paper does not achieve its goals. There are some suggestions I have that might improve the presentation. My specific comments are below.

**Response to General Comments:**

The authors thank Reviewer 1 (Christopher Spence) for the indispensable comments which have greatly aided in the clarity of the manuscript. The primary concern raised by Reviewer 1 relates to the dynamics of the soil frost routine. The authors recognize that the explicit nature of the modifications of the model to account for long-term freezing temperatures in water physics may not have been stated as clearly as necessary. For additional clarification, the authors will state that the model would not previously freeze water (regardless of temperature) as the model was not originally designed for cold regions. The modifications presented here allow the model to account for phase change of soil water during freezing conditions as well as limit the mobility of solid water.

**Major Comments**

**R1C1:** Page 1 Line 34: It is not clear how the limited number of monitoring sites is tied to implications of hydrological change. Maybe rephrase to "The limited number of long-term monitoring sites with high quality data is a concern because it may prove difficult to document the anticipated hydrological change in these catchments".

**Response to R1C1:** Thank you for the suggestions. The authors will revise the statement accordingly.

**R1C2:** Page 4 Line 39: How is the equation presented here related to the assumption that the ground and snowpack temperature are the same?

**Response to R1C2:** Within the model framework, the surface temperature $(T_s)$ is not directly isothermal with the snowpack, but is damped to imitate thermal conduction through a snowpack. The authors recognize that the earlier definition of the isothermal properties of temperature appeared to indicate that the whole soil profile was isothermal with the snowpack. This will be adjusted within the manuscript and explicitly state (after Page 4 Line 39) that a simple estimation of conduction is considered.

**R1C3:** Page 5 Line 50: Here and elsewhere, the paper would benefit greatly from the inclusion of units when introducing variables.

**Response to R1C3:** The authors will revise the text throughout the manuscript to ensure that the units of new variables are included.

**R1C4:** Page 5 Line 50: These equations imply the soil moisture scheme assumes no movement of water in the column? I cannot think this is correct, and I must misunderstand. Could the authors please improve the clarity here?

**Response to R1C4:** The reviewer is correct that this is not how water movement is estimated. Rather, the soil water is redistributed within EcH2O and the equation in the manuscript describes how the soil moisture changes after the redistribution due to freezing. This will be clarified in the revision.

**R1C5:** Equation 6: It might be the version I see, but the equation seems incomplete and the description doesn't quite match with no mention of outflow.

**Response to R1C5:** The authors thank the reviewer for noticing this typo. There were two subscripts missing in the equation and will be added in the revision.

**R1C6:** Page 5 Line 62: Perhaps show the equation from Ala-aho, to show the difference to the reader.

**Response to R1C6:** The authors thank reviewer 1 for the suggestion. The authors will include the equation (inline) to provide a direct comparison for how the modifications are conducted and the influence of the change, as well as the physical meaning of these changes.

**R1C7:** Figure 1 could be better drafted and explicitly label the locations of S12 and S22.

**Response to R1C7:** The authors will revise Figure 1 to include the locations of both S12 and S22 as an inset plot (inclusion within Figure 1a was previously tried but was too difficult to see S12 and S22).

**R1C8:** Page 6 Line 90: Not all of this section includes model data, and some is observational data. You could perhaps retitle the section "Observations".

**Response to R1C8:** The title of the section will be revised during revision

**R1C9:** Page 7 Line 20: Perhaps put the simulation period right at the beginning of the section.

**Response to R1C9:** The authors will move the simulation periods to the beginning of the section.

**R1C10:** Figure 2: Could the authors add a sentence or two explaining why the water ages bottom out every now and then? Perhaps I have missed it.

**Response to R1C10:** The water ages in the stream drop suddenly due to rain on snow events which result in rapid runoff of young water rapidly mixing with the older stream water. The relatively low streamflow volume during the winter months results in a large influence in the stream water ages. This will be added to the text.

**R1C11:** Page 10 Line 89: Are the words dynamic and damped mixed up?

**Response to R1C11:** The authors will revise this.

**R1C12:** Figure 3: Please explain what 'normalized' means.

**Response to R1C12:** The authors will add the description of normalizing to the manuscript figure to help make the figure stand-alone.

**R1C13:** Figure 3: Also, why does the soil water age get younger as the summer progress? The paper would benefit from a few sentences explaining this behaviour.

**Response to R1C13:** The soil water age decreased during the summer due to the flushing of older snowmelt and pre-winter water, replaced by younger growing season precipitation. Unlike many other studies, snowmelt age is accumulated throughout the winter months, therefore snowmelt is an aggregated age of the precipitation throughout the winter. The authors will add a statement explaining why the summer soil water ages decrease.

**R1C14:** Figure 5: Just so apples are compared to apples, perhaps total modelled evaporation and transpiration so that it can be more easily compared to the ICOS data.

**Response to R1C14:** The authors will add the ET to Figure 5d to help with the direct comparison of the measured evaporation and transpiration.

**R1C15:** Page 13 Line 41: A citation might be useful here because the data from this paper do not support such a statement.

**Response to R1C15:** The statement was conjecture and the authors will remove it from the manuscript.

**R1C16:** Page 15 Line 88: The authors have access to soil temperature data that could show if this is underestimated. A figure might help address this gap. Also, please explain how the assumption of no temperature gradient through the snowpack influence these results

**Response to R1C16:** The authors will clarify this statement. The modelled soil temperatures are not available throughout the soil domain but at two locations

**R1C17:** Page 15 Line 93 – 99: There are some typos through this section that could be fixed.

**Response to R1C17:** The authors will revise this section to improve the grammar to typos currently present.

**R1C18:** Page 16 Line 27: I missed where the ages of the soil frost are provided. It would be valuable to show them.

**Response to R1C18:** The authors will provide the ages of the soil frost in Figure 3 to aid the reviewer in understanding how the water ages are influenced by the soil frost estimations.

**R1C19:** Page 16 Line 30: It would be helpful to provide data on the relative values of these fluxes and storages in the text here to let the reader know how important each is to determine the age of water.

**Response to R1C19:** The authors thank the reviewer for the suggestion. Providing numerical values will aid the reader in understanding how the mixing and timing of different water sources will influence the water ages.

**R1C20:** Page 16 Line 32: Maybe rephrase to "...of older soil frost with younger soil water and snowmelt reduces....."

**Response to R1C20:** The authors will revise the statement using the reviewer's suggestion.

**R1C21:** Page 16 Line 35: Was it limited or just hard to detect within the uncertainties of the model? This is an important point of discussion that is missing.

**Response to R1C21:** Within the study site there is a combination effect. On some of the smaller streams, the influence of the soil-frost on the stream water age is more apparent, with younger snowmelt reaching the stream faster due to overland runoff as

with rain on snow (limited infiltration). With some model parameterization, the differences are more apparent; however, on average, these differences are minor and are not significant when compared to the model uncertainties of water ages. The authors will add this to the discussion section.

**R1C22:** Page 16 Line 43: I am not convinced the results of the research support these statements. Please clarify. If more water is pulled from soil subject to warming would not that speed up the pattern observed in Figure 3? And in turn reduce age?

**Response to R1C22:** The authors will revise this statement to clarify that the increase in water availability (regardless of vegetation) will result in higher water use of the younger water in the soils. Higher use of younger water for vegetation results in less young water feeding the groundwater and stream, thereby increasing the water ages.

---

## Author Comment (AC2) · 29 May 2019

**General Comments of Reviewer 2**

The paper by Smith et al., seeks to use a previously developed ecohydrological model (EcH2O-iso) to further understand the partitioning, water storage, flux and age interactions, particularly in the context of cold, northern catchments. This novelty of this contribution is that they have adapted the model to include soil freezing, and the impact of soil freezing on water ages. As the authors note, most model estimations of

storage-flux interactions oversimplify vegetation-soil-water interactions, while EcH2O-iso provides a generic and relatively simplistic (in some parts) modeling approach to evaluate storage and water ages in cold environments. The model of course has limitations related to the process physics and the assumption of complete isotope mixing within each compartment, which may not hold true. However, the authors are transparent as to its shortcomings in most places, and it is of little value to be overly picky with regards to the choices that are made. The manuscript is well written, and the figures are clear and of high quality. I would like the authors to consider the comments below and I believe the manuscript is suitable for publication after minor revisions. The main conclusion of the work is that soil frost had an early season influence on the ages of transpiration, with less of an influence on water ages of evaporation. Second, that the new module can simulate soil frost dynamics. While I do not dispute this, it is unsurprising that the Stefan-type of equations can simulate frost well, this approach has been used for ages and ages and while perhaps not always a physical realistic representation of ground freezing, it simply works well (as it does here). It would be good for the authors to indicate whey they did not use a more complex thermal scheme, or reference ones. Obviously, one would need more soil layers and computational resources would go through the roof, but a bit more on the 'why' this method was used is good. I would like to focus my comments around the central conclusion re: soil frost and water ages. It would be useful to outline how evaporation and transpiration are partitioned as this would help the reader (although it is likely presented elsewhere) and goes to the central conclusion.

**Response to General Comments of Reviewer 2**

The authors thank reviewer 2 for their constructive comments for the manuscript. The choice here for using the relatively simple Stefan's equation to solve for the soil frost depth because the authors were trying to minimize the changes to the model structure of EcH2O. The authors had explored more comprehensive thermal schemes, the current model structure of EcH2O resolves the energy balance at the surface through an

iterative approach and would not allow for a simple adaptation of the model structure. These changes would result in significant changes to how the energy balance of the surface (and also the canopy) is conducted. This would be an interesting development but would need additional testing in both the winter and summer conditions to ensure that there is not a significant error with a different energy balance estimation. The authors will separately consider how the evaporation and transpiration are estimations in the model description.

**Major Comments**

**R2C1:** Equation 1 simulates the depth of the freezing front, but not the soil temperature. I am curious as to how the model simulates soil temperature. I THINK I understand how the surface temperature is driven, and the authors acknowledge that the thermal routine of the snowpack is simple for various reasons. What I'm trying to get at is: does the model simulate a soil temperature and how does this relate to the position of the zero-degree isotherm. Yes, soils will be identified as frozen or unfrozen base on Eq1, yet is there a modeled soil temperature that simply has no freezing routine? More clarity is needed.

**Response to R2C1:** The model does estimate a soil temperature; however, the implementation of soil temperature was derived for warm climates where there are not discontinuities with the thermal conductivity or heat capacity (due to ice conditions). These discontinuities influence the depth that the soil temperature is recorded as there is no soil temperature profile estimated within the model framework so it is not currently possible to identify the zero-degree isotherm in the model. There is additional work to be conducted with the model to account for the ground heat flux under the snowpack, which may require a more robust thermal diffusion through the snowpack to properly estimate the soil temperatures. The authors will clarify these points in the manuscript when introducing the energy balance of EcH2O.

**R2C2:** The central conclusion that soil freezing affects transpiration is fine, but is it

simply because the plants are not 'on' when the soil is frozen and soil evaporation is impeded (it certainly would be). When the module is off, plants can transpire, and soils evaporate? Is it this simple? I'm just not sure. More clarity on what drives the transpiration would be helpful as I'm unsure if there can be no transpiration when the rooting zone is frozen – how does this all work?

**Response to R2C2:** The authors will clarify the effect of frost on transpiration and evaporation estimation in the discussion section. The soil frost does not turn 'on' or 'off' the transpiration or soil evaporation, rather, the soil frost restricts the water available for transpiration. As transpiration is a function of the water available, the transpiration is reduced. When the soil frost routine is 'off', more water is available for the transpiration and thereby isn't restricted to the same degree.

**R2C3:** Is there sublimation in the model? I see that latent heat is set at 0 when there is snow – why? What impact does this have when snow is melting and sublimation maybe important.

**Response to R2C3:** There is not currently a sublimation module from the snowpack surface within EcH2O (latent heat set to 0). However, there is some sublimation from the canopy interception, though there is not currently a phase dependent interception within the model (SWE and rainfall depths have the same interception capacity in vegetation). This may have some effect on the water volumes during the spring months and increase the dominance of the soil frost conditions. A brief discussion will be included about the influence of sublimation.

**R2C4:** For Equation 7, what is the basis of the amplification factor C. Does equation 7 pre-serve an isotope mass balance throughout all time steps (I'm assuming so – but it should be stated).

**Response to R2C4:** The meaning of C will be added to the manuscript. While S is representative of the shape of the snowmelt fractionation curve (i.e. timing of the melt), the amplification factor is representative of the atmospheric effect on the fractionation

(i.e. RH and temperature effect). To minimize "moving parts" in the model, this was held constant and calibrated. The manuscript will be adjusted to better reflect what this term means and that Eq 7 serves as the isotope mass for the snowpack of the model.

**R2C5:** The authors use ERA-Interim data to drive the radiative component of the model. For a few years, there was overlap. Did they investigate the bias of the ERA data and correct? I'm assuming ERA-I would work well in this location of Europe, but it's good to check as it can have biases which will propagate through the energy balance calculations. The underestimation in net radiation is a bit concerning – and latent heat as well. So after all this, my question is that if latent heat is, in fact, greater than simulated, what influence would this have on the age estimates (if any?). I assume some and this should be noted.

**Response to R2C5:** The authors will discuss the influence of the use of the ERA-data within the study and how the use of different radiation forcing data may impact the results of water ages. The authors did examine the difference between the ERA data to the on-site data and there is no noticeable bias with the ERA data at the site.

**R2C6:** On line 79, I'm not sure that the CRHM reference is correct and the Xie and Gough paper describes the thermal routine that is later incorporated into CRHM (see papers by Krogh for example). The XG method is in CRHM, but this is just slightly incorrect referencing.

**Response to R2C6:** The authors will revise this reference.

**R2C7:** The discussion after line 85 is a bit selective and there are dozens of possible reasons for model errors in turbulent fluxes. First, the authors state sensible heat fluxes are underestimated but only show latent fluxes so the discussion should be there or sensible heat data should be provided. Another reason not stated (and noted above) is the nature of the ERA-I data. I'm also unsure as to how snow processes are incorporated into the canopy module re: unloading, albedo change, etc. All I'm saying is that there are many many reasons here where the model could be improved with

physics, and avoid suggesting 'direct calibration' is the best way to improve simulations.

**Response to R2C7:** The sensible and latent heat fluxes (with the net radiation) are all provided in Figure 5 (rather than just latent heat). It was not the intention of the authors to state that "direct calibration" was the only means to improve the estimation of the heat flux estimation, rather than some of the model parameters may be sensitive to the heat flux that was not included in the calibration because the heat flux (evaporation or transpiration either for that matter) were not calibrated. Thereby, any inclusion of those parameters would yield no significant posterior distribution. The authors will modify this section to indicate that there are multiple processes (some of which are not yet included in the model) that may influence the energy balance while stating that the potential reasons are examples of influences.

**R2C8:** Figures that highlight the differences between soil moisture at depth would be helpful.

**Response to R2C8:** The authors chose not to include the soil moisture at depth since the model was not directly calibrated to different soil moisture depths. The authors believe that showing this dataset may result in confusion that the model was calibrated to distinct soil depths. This calibration was not directly possible due to the calibration of soil layer depths 1 and 2 which limited the comparability of the calibration with variable depths. The soil moisture at different depths will be added to the appendix for additional information for the readers.

**Specific Comments**

**R2C9:** Line 80: under different vegetation communities (forest vs mire). 2) To examine the influence of soil frost on the dynamics and age of water (Comma instead of a period after(forest vs mire)

**Response to R2C9:** The authors will revise this typo.

**R2C10:** Line 54: qin: subscript needs to be added

**Response to R2C10:** The authors will add the subscripts.

**R2C11:** Line 73: comma needed within coordinates

**Response to R2C11:** The authors will revise the statement to include the comma.

**R2C12:** Line 95: "Stable isotopes determinations were carried out" : Fix wording

**Response to R2C12:** The authors will revise this wording.

**R2C13:** Table 1: Units of precipitation say m/s → should be moved to wind speed. Units need to be added to other dat. "30 min for Sensible Heat says " 30 in" . Column heading needs to say "Time Period" for top row.

**Response to R2C13:** The authors will revise this table to include units for all input and calibration /validation data.

**R2C14:** Line 69: stream isotopes tended to retain a slight "memory" effect from the more enriched late summer..."contributions"? "water"? I think a word is needed here?

**Response to R2C14:** The authors will revise this statement.

**R2C15:** Beginning Line 95: While some work has been conducted on assessing the transit or residence times of ecohydrologic fluxes or their partitioning in northern (e.g. Sprenger et al., 2018a); however, few studies have included the influence of frozen conditions on the water movement, which may be significant for the effective transit times during the spring freshet period (Tetzlaff et al., 2018) and flow path modelling in "cold" regions(Laudon et al., 2007; Sterte et al., 2018).

**Response to R2C15:** The authors will revise this statement to improve clarity.

**R2C16:** Line 99: Traditionally, water ages in stream water at catchment outlets have been the primary metrics for assessing the transport of tracers. Should this read: Traditionally, isotopic tracers in stream water at catchment outlets have been the primary metrics for assessing water ages.

**Response to R2C16:** The authors will revise this statement with the reviewer's suggestion.

**R2C17:** Line 29: snow and early spring snowmelt), and snowpack is the amount "weighed" age of solid precipitation (*Should this be weighted)

**Response to R2C17:** The authors will revise this statement with the reviewer's suggestion.

---

## Author Response (AR1)

**General Comments R1**

In this paper, the authors apply a tracer aided hydroecological model to assess the role of frozen ground on water fluxes, storage and ages in a cold regions watershed in northern Sweden. The model performed well enough to make sound conclusions about the relative magnitude of fluxes and the distribution of ages of water comprising different components of the water budget. The subject matter of this research is very relevant in regards to beginning to address larger questions about how climate, vegetation and hydrology interact. These are important questions as the globe warms, and tools such as the model introduced here will be important for predicting and attributing change. The paper is well written. I have some minor suggestions where improvements could be made. A bigger concern is an incomplete explanation of how the authors assessed the role of ground frost on water fluxes and ages. The authors explain that they turn frost dynamics off in the model to do so. I perhaps misunderstand, but how is it possible to not have the soil freeze if the same forcing dataset is used? This is a crucial piece in the methodology and it needs better explaining than currently exists. Without it, the paper does not achieve its goals. There are some suggestions I have that might improve the presentation. My specific comments are below.

**Response to General Comments R1:**

The authors thank Reviewer 1 (Chris Spence) for the indispensable comments which have greatly aided in the clarity of the manuscript. The primary concern raised by Reviewer 1 relates to the dynamics of the soil frost routine. The authors recognize that the explicit nature of the modifications of the model to account for long-term freezing temperatures in water physics may not have been stated as clearly as necessary. For additional clarification, the authors have revised the manuscript to state that the model would not previously freeze water (regardless of temperature) as the model was not originally designed for cold regions. The modifications presented here allow the model to account for phase change of soil water during freezing conditions as well as limit the mobility of solid water.

**Major Comments**

**R1C1:** Page 1 Line 34: It is not clear how the limited number of monitoring sites is tied to implications of hydrological change. Maybe rephrase to "The limited number of long-term monitoring sites with high quality data is a concern because it may prove difficult to document the anticipated hydrological change in these catchments".

**Response to R1C1:** The authors thank reviewer 1 for the suggestion, and have revised the statements (Lines 33-35, Page 1)

**R1C2:** Page 4 Line 39: How is the equation presented here related to the assumption that the ground and snowpack temperature are the same?

**Response to R1C2:** Within the model framework implemented with the freeze-thaw cycles, the surface temperature below the snowpack ($T_s$) is not isothermal with the snowpack or the temperature above the snowpack. Estimations of the frost depth using isothermal estimations would result in an overestimation of the frost depth. To clarify that the soil profile and snowpack is are not isothermal the authors have adjusted the manuscript to better explain the temperature damping and thermal conduction (Lines 151 – 154, Pages 4-5).

**R1C3:** Page 5 Line 50: Here and elsewhere, the paper would benefit greatly from the inclusion of units when introducing variables.
**Response to R1C3:** The authors have revised the manuscript to include units of variables where they are introduced (e.g. Line 167, Page 5).

**R1C4:** Page 5 Line 50: These equations imply the soil moisture scheme assumes no movement of water in the column? I cannot think this is correct, and I must misunderstand. Could the authors please improve the clarity here?
**Response to R1C4:** The reviewer is correct that this is not how water movement is estimated. The change in soil water due to freezing shown with Eqn. 5 occurs after water redistributed within EcH2O. The authors have clarified the statement before the equation to indicate that redistribution occurs before the estimation of soil moisture change due to freezing (Lines 165-166, page 5).

**R1C5:** Equation 6: It might be the version I see, but the equation seems incomplete and the description doesn't quite match with no mention of outflow.
**Response to R1C5:** The authors thank the reviewer for noticing this typo. There were two subscripts missing in the equation and have been added in the revision (Eqn 6. Line 170, Page 5).

**R1C6:** Page 5 Line 62: Perhaps show the equation from Ala-aho, to show the difference to the reader.
**Response to R1C6:** The authors thank reviewer 1 for the suggestion. The authors have included the equation (inline, Line 182, Page 5) to provide a direct comparison for how the modifications are conducted and the influence of the change, as well as the physical meaning of these changes.

**R1C7:** Figure 1 could be better drafted and explicitly label the locations of S12 and S22.
**Response to R1C7:** The authors have revised Figure 1 to include the locations of both S12 and S22 as an inset plot.

**R1C8:** Page 6 Line 90: Not all of this section includes model data, and some is observational data. You could perhaps retitle the section "Observations".
**Response to R1C8:** The authors have revised the title of the section to "Input and calibration datasets" (Line 213, Page 7).

**R1C9:** Page 7 Line 20: Perhaps put the simulation period right at the beginning of the section.

**Response to R1C9:** The authors have moved the statement of the periods of the simulation to the beginning of the section (Lines 241 – 243, Page 8).

**R1C10:** Figure 2: Could the authors add a sentence or two explaining why the water ages bottom out every now and then? Perhaps I have missed it.
**Response to R1C10:** The water ages in the stream drop suddenly due to rain on snow events which result in rapid runoff of young water rapidly mixing with the older stream water. The relatively low streamflow volume during the winter months results in a large influence in the stream water ages. This explanation has been added to the manuscript (Lines 301 – 303, Page 10).

**R1C11:** Page 10 Line 89: Are the words dynamic and damped mixed up?
**Response to R1C11:** The authors have revised the wording (Lines 313 – 314, Page 11).

**R1C12:** Figure 3: Please explain what 'normalized' means.
**Response to R1C12:** The authors have added the description to the manuscript (Line 268, Page 9) and within the figure (Line 325, Page 10).

**R1C13:** Figure 3: Also, why does the soil water age get younger as the summer progress? The paper would benefit from a few sentences explaining this behaviour.
**Response to R1C13:** The soil water age decreased during the summer due to the flushing of older snowmelt and pre-winter water, replaced by younger growing season precipitation. Unlike many other studies, snowmelt age is accumulated throughout the winter months, and snowmelt is not input to catchment storage with an age of 0 days. The authors have added a statement to explain the decrease in soil water ages during the summer (Lines 319 – 320, Page 11).

**R1C14:** Figure 5: Just so apples are compared to apples, perhaps total modelled evaporation and transpiration so that it can be more easily compared to the ICOS data.
**Response to R1C14:** The authors have added the simulated ET to Figure 5d to directly compare to the ICOS tower ET (Fig. 5, Page 13).

**R1C15:** Page 13 Line 41: A citation might be useful here because the data from this paper do not support such a statement.
**Response to R1C15:** The authors have removed the statement from the manuscript.

**R1C16:** Page 15 Line 88: The authors have access to soil temperature data that could show if this is underestimated. A figure might help address this gap. Also, please explain how the assumption of no temperature gradient through the snowpack influence these results
**Response to R1C16:** The authors have clarified in the manuscript (Line 115, Page 3) that the soil temperature is only estimated at one location in the soil domain (the interface of the first and second thermal layers). Similar to Response to R1C2, there is a diffusive gradient of temperature through the snowpack estimated with a calibration parameter (Line 154, Page 5). The authors have also added some discussion on potential over-estimation of heat sink (under-estimation of thermal conduction) to the discussion (Lines 416 – 418, Page 15).

**R1C17:** Page 15 Line 93 – 99: There are some typos through this section that could be fixed.
**Response to R1C17:** The authors have revised this section to improve the grammar and typos (Lines 424 – 430, page 15) .

**R1C18:** Page 16 Line 27: I missed where the ages of the soil frost are provided. It would be valuable to show them.
**Response to R1C18:** The authors have revised Figure 3 to include the water ages of soil frost (Page 11).

**R1C19:** Page 16 Line 30: It would be helpful to provide data on the relative values of these fluxes and storages in the text here to let the reader know how important each is to determining the age of water.
**Response to R1C19:** The authors thank the reviewer for the suggestion. The authors have provided numerical values in-line to aid the reader in understanding how the mixing and timing of different water sources will influence the water ages (Lines 468 – 469, Page 16).

**R1C20:** Page 16 Line 32: Maybe rephrase to "...of older soil frost with younger soil water and snowmelt reduces....."
**Response to R1C20:** The authors have revised the statement using the reviewer's suggestion (Lines 470 – 471, Page 16).

**R1C21:** Page 16 Line 35: Was it limited or just hard to detect within the uncertainties of the model? This is an important point of discussion that is missing.
**Response to R1C21:** There is a combined effect of model uncertainty and limited effect. On the smaller streams, the influence of the soil-frost on the stream water age is more apparent, with younger snowmelt reaching the stream faster due to overland runoff as with rain on snow (limited infiltration). In some model parameterizations the differences are apparent in all streams; however, on average, these differences are not significant when compared to the model uncertainties of water ages. The authors have added this to the discussion section (Lines .432 – 436, Page 15).

**R1C22:** Page 16 Line 43: I am not convinced the results of the research support these statements. Please clarify. If more water is pulled from soil subject to warming would not that speed up the pattern observed in Figure 3? And in turn reduce age?

**Response to R1C22:** The authors have revised this statement to clarify that the increase in water availability (regardless of vegetation) will result in higher water use of the younger water in the soils. Higher use of younger water for vegetation results in less young water feeding the groundwater and stream, thereby increasing the water ages (Lines 478 – 482, Page 16).

**General Comments of Reviewer 2**

The paper by Smith et al., seeks to use a previously developed ecohydrological model (EcH2O-iso) to further understand the partitioning, water storage, flux and age inter-actions, particularly in the context of cold, northern catchments. This novelty of this contribution is that they have adapted the model to include soil freezing, and the impact of soil freezing on water ages. As the authors note, most model estimations of storage-flux interactions oversimplify vegetation-soil-water interactions, while EcH2O-iso provides a generic and relatively simplistic (in some parts) modeling approach to evaluate storage and water ages in cold environments. The model of course has limitations related to the process physics and the assumption of complete isotope mixing within each compartment, which may not hold true. However, the authors are transparent as to its shortcomings in most places, and it is of little value to be overly picky with regards to the choices that are made. The manuscript is well written, and the figures are clear and of high quality. I would like the authors to consider the comments below and I believe the manuscript is suitable for publication after minor revisions. The main conclusion of the work is that soil frost had an early season influence on the ages of transpiration, with less of an influence on water ages of evaporation. Second, that the new module can simulate soil frost dynamics. While I do not dispute this, it is unsurprising that the Stefan-type of equations can simulate frost well, this approach has been used for ages and ages and while perhaps not always a physical realistic representation of ground freezing, it simply works well (as it does here). It would be good for the authors to indicate whey they did not use a more complex thermal scheme, or reference ones. Obviously, one would need more soil layers and computational resources would go through the roof, but a bit more on the 'why' this method was used is good. I would like to focus my comments around the central conclusion re: soil frost and water ages. It would be useful to outline how evaporation and transpiration are partitioned as this would help the reader (although it is likely presented elsewhere) and goes to the central conclusion.

**Response to General Comments of Reviewer 2**

The authors thank reviewer 2 for their constructive comments for the manuscript. The choice here for using the relatively simple Stefan's equation to solve for the soil frost depth because the authors were trying to minimize the changes to the model structure of EcH2O. The authors had explored more comprehensive thermal schemes, the current model structure of EcH2O resolves the energy balance at the surface through an iterative approach and would not allow for a simple adaptation of the model structure. Changes to the model structure would significant alter the energy balance of the surface (and also the canopy). This would be an interesting development but would need additional testing in both the winter and summer conditions to ensure that there is not a significant error with a different energy balance estimation. Additional estimation methods of soil frost have been added to the introduction section. The authors have added further descriptions of how the evaporation and transpiration are estimated in the model description section.

*Major Comments*

**R2C1:** Equation 1 simulates the depth of the freezing front, but not the soil temperature. I am curious as to how the model simulates soil temperature. I THINK I understand how the surface temperature is driven, and the authors acknowledge that the thermal routine of the snowpack is simple for various reasons. What I'm trying to get at is: does the model simulate a soil temperature and how does this relate to the position of the zero-degree isotherm. Yes, soils will be identified as frozen or unfrozen base on Eq1, yet is there a modeled soil temperature that simply has no freezing routine? More clarity is needed.

**Response to R2C1:** The model does estimate a soil temperature; however, the implementation of soil temperature was derived for warm climates where there are not discontinuities with the thermal conductivity or heat capacity (due to ice conditions). These discontinuities influence the depth that the soil temperature is recorded as there is no soil temperature profile estimated within the model framework so it is not currently possible to identify the zero-degree isotherm in the model. There is additional work to be conducted with the model to account for the ground heat flux under the snowpack, with may require a more robust thermal diffusion through the snowpack to properly estimate the soil temperatures. The authors have clarified the estimation of the soil temperatures when introducing the energy balance of EcH$_2$O (Lines 112- 117, Page 3).

**R2C2:** The central conclusion that soil freezing affects transpiration is fine, but is it simply because the plants are not 'on' when the soil is frozen and soil evaporation is impeded (it certainly would be). When the module is off, plants can transpire, and soils evaporate? Is it this simple? I'm just not sure. More clarity on what drives the transpiration would be helpful as I'm unsure if there can be no transpiration when the rooting zone is frozen – how does this all work?

**Response to R2C2:** The authors have clarified this in the discussion section. The soil frost does not turn 'on' or 'off' the transpiration or soil evaporation, rather, the soil frost restricts the water available for transpiration. As transpiration is a function of the water available, the transpiration is reduced. When the soil frost routine is 'off', more water is available for the transpiration and thereby isn't restricted to the same degree (Lines 458 – 461, Page 16).

**R2C3:** Is there sublimation in the model? I see that latent heat is set at 0 when there is snow – why? What impact does this have when snow is melting and sublimation maybe important.

**Response to R2C3:** There is not currently a sublimation module from the snowpack surface within EcH$_2$O (latent heat set to 0). However, there is some sublimation from the canopy interception, though there is not currently a phase dependent interception within the model (SWE and rainfall depths have the same interception capacity in vegetation). This may have some effect on the water volumes during the spring months and increase the dominance of the soil frost conditions. A brief section to the discussion section has been added about the influence of sublimation (Lines 414 – 422, page 15).

**R2C4:** For Equation 7, what is the basis of the amplification factor C. Does equation 7 pre-serve an isotope mass balance throughout all time steps (I'm assuming so – but it should be stated).
**Response to R2C4:** The authors recognize that the meaning of C was not provided in the manuscript. While S is representative of the shape of the snowmelt fractionation curve (i.e. timing of the melt), the amplification factor is representative of the atmospheric effect on the fractionation (i.e. RH and temperature effect). To minimize "moving parts" in the model, this was held constant and calibrated. The manuscript has been revised to define C and describe the definition of Eq 7 (Lines 187 – 193, Page 6)

**R2C5:** The authors use ERA-Interim data to drive the radiative component of the model. For a few years, there was overlap. Did they investigate the bias of the ERA data and correct? I'm assuming ERA-I would work well in this location of Europe, but it's good to check as it can have biases which will propagate through the energy balance calculations. The underestimation in net radiation is a bit concerning – and latent heat as well. So after all this, my question is that if latent heat is in fact greater than simulated, what influence would this have on the age estimates (if any?). I assume some and this should be noted.
**Response to R2C5:** The authors have added a discussion on the influence of the use of the ERA-data within the study and how this may influence the results in the study. The authors did examine the difference between the ERA data to the on-site data and there is no noticeable bias with the ERA data at the site (Lines 418 – 422, Page 15)

**R2C6:** On line 79, I'm not sure that the CRHM reference is correct and the Xie and Gough paper describes the thermal routine that is later incorporated into CRHM (see papers by Krogh for example). The XG method is in CRHM, but this is just slightly incorrect referencing.
**Response to R2C6:** The authors have revised this reference (Line 403, Page 14).

**R2C7:** The discussion after line 85 is a bit selective and there are dozens of possible reasons for model errors in turbulent fluxes. First the authors state sensible heat fluxes are underestimated but only show latent fluxes so the discussion should be there or sensible heat data should be provided. Another reason not stated (and noted above) is the nature of the ERA-I data. I'm also unsure as to how snow processes are incorporated into the canopy module re: unloading, albedo change, etc. All I'm saying is that there are many many reasons here where the model could be improved with physics, and avoid suggesting 'direct calibration' is the best way to improve simulations.
**Response to R2C7:** The sensible and latent heat fluxes (with the net radiation) are all provided on Figure 5 (rather than just latent heat). It was not the intention of the authors to state that "direct calibration" was the only means to improve the estimation of the heat flux estimation, rather that some of the model parameters may be sensitive to the heat flux that were not included in the calibration because the heat flux (evaporation or transpiration either for that matter) were not calibrated. Thereby, any inclusion of those parameters would yield not significant posterior distribution. The authors have modified this section to indicate that there are multiple processes (some of which are not yet included in the model) that may influence the energy balance while stating that the potential reasons are examples of influences (Lines 411 – 422, Page 15).

**R2C8:** Figures that highlight the differences between soil moisture at depth would be helpful.
**Response to R2C8:** The authors chose to not include the soil moisture at depth since the model was not directly calibrated to different soil moisture depths, and showing this dataset may result in confusion that the model was calibrated to distinct soil depths. This calibration was not directly possible due to the calibration of soil layer depths 1 and 2 which limited the comparability of the calibration with variable depths. The soil moisture at different depths have been added to the appendix for additional information for the readers.

**Specific Comments**
**R2C9:** Line 80: under different vegetation communities (forest vs mire). 2) To examine the influence of soil frost on the dynamics and age of water (Comma instead of period after(forest vs mire)
**Response to R2C9:** The authors have revised this typo (Line 85, Page 3)..

**R2C10:** Line 54: qin → subscript needs to be added
**Response to R2C10:** The authors have revised this typo (Eq 6, Page 5).

**R2C11:** Line 73: comma needed within coordinates
**Response to R2C11:** The authors have revised this typo (Line 196, Page 6).

**R2C12:** Line 95: "Stable isotopes determinations were carried out" → Fix wording
**Response to R2C12:** The authors have revised this wording (Line 218, Page 7).

**R2C13:** Table 1: Units of precipitation say m/s → should be moved to wind speed. Units need to be added to other dat. "30 min for Sensible Heat says " 30 in" . Column heading needs to say "Time Period" for top row.
**Response to R2C13:** The authors have revised this table to include units for all input and calibration /validation data (Table 1, Page 7).

**R2C14:** Line 69: stream isotopes tended to retain a slight "memory" effect from the more enriched late summer..."contributions"? "water"? I think a word is needed here?
**Response to R2C14:** The authors have revised this statement (Line 292, Page 10).

**R2C15:** Beginning Line 95: While some work has been conducted on assessing the transit or residence times of ecohydrologic fluxes or their partitioning in northern (e.g. Sprenger et al., 2018a); however, few studies have included the influence of frozen conditions on the water movement, which may be significant for the effective transit times during the spring freshet period (Tetzlaff et al., 2018) and flow path modelling in "cold" regions(Laudon et al., 2007; Sterte et al., 2018).

**Response to R2C15:** The authors have revised this statement (lines 427 – 430, Page 15).

[revised manuscript text omitted]